# LEARNING THE LATENT NOISY DATA GENERATIVE PROCESS FOR LABEL-NOISE LEARNING

## ABSTRACT

In learning with noisy labels, the noise transition reveals how an instance relates from its clean label to its noisy one. Accurately inferring an instance's noise transition is crucial for inferring its clean label. However, when only a noisy dataset is available, noise transitions can typically be inferred only for a "special" group of instances. To use these learned transitions to assist in inferring others, it is essential to understand the connections among different transitions across different instances. Existing work usually addresses this by introducing assumptions that explicitly define the similarity of noise transitions across various instances. However, these similarity-based assumptions often lack empirical validation and may not be aligned with real-world data. The misalignment can lead to misinterpretations of both noise transitions and clean labels. In this work, instead of directly defining similarity, we propose modeling the generative process of noisy data. Intuitively, to understand the connections among noise transitions across different instances, we represent the causal generative process of noisy data using a learnable graphical model. Relying solely on noisy data, our method can effectively discern the underlying causal generative process, subsequently inferring the noise transitions of instances and their clean labels. Experiments on various datasets with different types of label noise further demonstrate our method's effectiveness.

## 1 INTRODUCTION

Supervised learning relies on annotated large-scale datasets, which can be both time-consuming and costly to create. Although several existing annotation methods offer cost-effective alternatives, such as online queries (Blum et al., 2003), crowdsourcing (Yan et al., 2014), and image engines (Li et al., 2017), the datasets obtained by these methods are imperfect. The labels of these datasets usually contain errors. These noisy labels would be harmful to deep neural networks because the network can memorize noisy labels easily (Zhang et al., 2017; Han et al., 2018; Bai et al., 2021) and lead to the degeneration of classification accuracy.

Modeling the noise transition plays an important role in many label-noise learning algorithms (Liu & Tao, 2016; Patrini et al., 2017; Xia et al., 2019; Li et al., 2021). Let $Y$, $\boldsymbol{X}$ and $\tilde{Y}$ denote the variables of the clean label, instance and noisy label, respectively. The noise transition for an instance $\boldsymbol{x}$ can be represented by $P(\tilde{Y}|Y, X = \boldsymbol{x})$, which reveals the probability distribution of the event that given an instance, its latent clean label is transited to the observed noisy label. If the noise transition is given, classifiers learned on noisy data can be used to infer the optimal ones defined by the clean data, with theoretical guarantees (Liu & Tao, 2016; Patrini et al., 2017; Xia et al., 2019).

However, noise transitions are generally unknown and need to be inferred. When given only a noisy dataset, noise transitions can be inferred for a "special" group of instances. For example, if instances belong to a clean class with probability one, their noise transition can be inferred (Xia et al., 2019). To use these learned transitions to assist in inferring others, it is essential to understand the connections among different transitions across different instances. Prior work tackles this issue by making additional assumptions. They manually define a similarity for noise transition across various instances. For example: (1). The noise transition is class dependent, which means that the noise transition for the instances in a class is the same (Liu & Tao, 2016; Patrini et al., 2017; Xia et al., 2019); (2). The noise transition for the instances in the same manifold is the same (Cheng et al., 2022a); (3). For the instances and their two nearest-neighbor instances belong to the same true class,

these instances share a same noise transition (Zhu et al., 2021). However, the similarity in previous work is defined by human prior knowledge. It is unclear whether these predefined similarities are truthfulness to the underlying connection. Given only noisy data, they are hard to verify. If similarities are not truthful, the estimation error of the noise transition could be large.

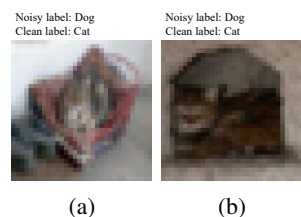

In this paper, to understand the connections of noise transitions among different instances without predefined similarity, we propose to explore the generative process of noisy data. By understanding this process, the connection of noise transitions across instances in different datasets would be automatically captured. Consider the CIFAR-10N dataset, shown in Fig. 1. The annotator, with a keen eye for animal features like furs, mislabels both cat images as "dog" due to their furry appearance. Recognizing that the noise transition of the first image is from "cat" to "dog" allows us to extrapolate this transition to the second image. This extrapolation is possible because both images share a common causal factor furs that influence their labeling. In essence, the connections between noise transitions across different instances are established through these causal factors.

Figure 1: The pictures contain the same noisy labels.

The heart of our exploration lies in understanding the generative process of noisy data. This process is anchored by causal factors, which significantly influence how noise transitions manifest across instances. Drawing from the aforementioned example, if we can decode the generative process and pinpoint "furs" as a causal factor for generating noisy labels, then two images containing this factor will likely have similar noise transitions. Such insights enable us to establish connections between the noise transitions of various images.

However, a challenge emerges: the causal factors responsible for generating the noisy data are latent. Similarly, the generative process, which hinges on these causal factors, also remains latent. Despite our extensive search, we found no existing methods that attempt to unveil these latent noisy data generative processes. To address this void, we introduce a novel generative model that aims to infer the latent generative processes from noisy labels. To achieve this, we use encoders to infer latent causal factors and clean labels from noisy data. Then model the generative process by a learnable graph. Both empirical evidence and theoretical analysis demonstrate that our method can effectively determine the underlying causal generative process, which in tune helps to learn clean labels.

The rest of this paper is organized as follows. Section 2 provides a brief overview of related work in the field. Section 3 presents our proposed method and its implementation. In Section 4, we conduct experiments using synthetic and real-world datasets to validate the effectiveness of the proposed method. Finally, in Section 5, we summarize our paper.

## 2 RELATED WORK

In this section, we briefly review the related literature.

**Noise Transition Modeling** Modeling noise transition is important in many algorithms to learn with noisy labels. With the provided noise transition, the optimal classifier defined on the clean data can be learned using infinite noisy data. The philosophy is that the clean class posterior can be inferred using noisy class posterior and noise transition. Many previous works demonstrate that the noise transition can be estimated based on some training instances. For example, the noise transition can be estimated through the *anchor points* whose clean class posterior of a class is one (Liu & Tao, 2016; Patrini et al., 2017; Xia et al., 2019); The noise transition can be estimated through a set of examples with theoretically guaranteed Bayes optimal labels (Yang et al., 2022). To use the estimated noise transitions on other instances, previous work with theoretical guarantee defines the similarity of noise transitions across instances. For example, the noise transition is class-dependent (Liu & Tao, 2016; Patrini et al., 2017; Xia et al., 2019); Xia et al. (2020) propose that the transition matrices are dependent on image parts.

**Other Methods in Learning with Noisy Labels** Some noise-robust algorithms (Han et al., 2018; Li et al., 2020) select examples deemed likely to be accurate for training purposes. These selections are based on the memorization effect (Zhang et al., 2017; Liu et al., 2020; Bai et al., 2021), which

suggests deep neural networks initially learn dominant patterns before progressively learning less common ones. In noisy label environments, accurate labels often constitute the majority, leading networks to prioritize learning from examples with accurate labels, typically indicated by lower loss values. Co-Teaching (Han et al., 2018) employs this principle to identify low-loss examples as likely accurate. DivideMix (Li et al., 2020) uses a Gaussian Mixture Model to separate training examples into labeled and unlabeled sets based on their training loss, with the labeled set presumed to contain accurate labels. Additionally, some methods use generative models to facilitate learning with noisy labels. CausalNL (Yao et al., 2021) and InstanceGM (Garg et al., 2023) utilize instance-specific information to enhance classifier learning. Conversely, NPC (Bae et al., 2022) focuses on the generative process of estimated clean labels, not the noisy data generation, using generative models for label calibration. Finally, SOP (Liu et al., 2022a) applies the *sparse* property of the label noise, *i.e.*, incorrect labels are the minority, to prevent models from overfitting to label noise.

**Identifiable and Causal Representation Learning** Representation learning aims to identify the underlying latent variables that generate the observed low-dimensional variables. The mapping from latent variables to the observed variables is usually linear. However, previous studies prove that it is ill-posed to recover the latent variables with only these observed variables (Hyvärinen & Pajunen, 1999; Locatello et al., 2019). Recent advances in non-linear independent component analysis (ICA) show that it is possible to identify latent variables using auxiliary information. For example, using the temporal information to identify the latent variables (Sprekeler et al., 2014; Hyvärinen & Morioka, 2016; 2017); auxiliary variables that modulate the distribution of latent variables can be accessed (Hyvärinen et al., 2019; Khemakhem et al., 2020). Causal representation learning (Schölkopf et al., 2021) aims to recover the latent causal factors and causal structure from observed low-dimensional variables. The definition of causal factor is more strict. The change in some factors can influence other factors. Yang et al. (2021) proposes leveraging the auxiliary labels related to causal factors to recover the latent causal factors and structure. Brehmer et al. (2022) learn the causal representations by using the paired samples before and after interventions. Lachapelle et al. (2022) learn the causal representations by regularizing the latent causal structure to be sparse. Causal representations can be learned from time-series data with the observed intervention targets (Lippe et al., 2022; 2023).

## 3 LEARNING THE LATENT NOISY DATA GENERATIVE PROCESS

Understanding the generative process is critical to understanding the connection of noise transitions across different instances. We propose a practical method to learn the generative process, inspired by the latest theoretical advances in latent causal representation learning (Yang et al., 2021; Liu et al., 2022b). Emerging theories in causality (Hyvärinen et al., 2019; Khemakhem et al., 2020; Yang et al., 2021; Liu et al., 2022b) suggest that we can efficiently infer the latent generative process with the aid of additional supervised information. However, in learning with noisy labels, this additional supervised information necessitates clean labels. Fortunately, the task of selecting clean examples from noisy training data is well-established and well-studied. Many current techniques ensure effective selection, and some are even backed by theoretical guarantees.

We propose to use the supervision information on the selected clean examples to learn a classification network. This network is used to infer the clean label for each instance. Then, an encoder and two decoder, which are used to capture the generative process, can be learned by using the Variational AutoEncoder framework. The trained encoder and decoders can be used to regularize the classification network and improve the classification performance.

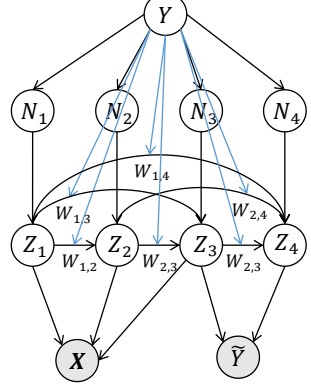

Figure 2: A illustration of the noise transition generative model with 4 latent variables.

**Problem Setting** Let $X$ and $\tilde{Y}$ denote the observed variables of instances and noisy labels, respectively. The observed variables $X$ and $\tilde{Y}$ are influenced by the causal factors $Z$. The causal factors are causally related, which means that some variables are the effects of other variables. The effects of other variables to a causal factor $Z_i$ can be represented as $Z_i := f_Z(pa(Z_i), N_i)$, where $f_Z(\cdot)$ represents

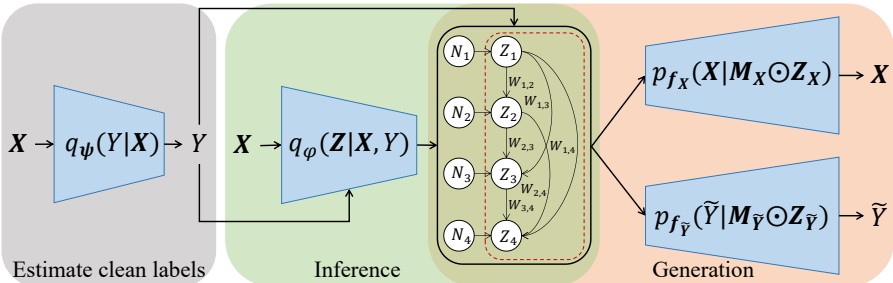

Figure 3: A working flow of our method. In the clean labels estimation stage, a classification network is used to infer the clean labels of instances; In the inference stage, an instance encoder and a noisy label encoder are used to infer the causal factors; In the generation stage, a decoder is used to generate the instances and noisy labels.

the causal mechanism, $pa(Z_i)$ represents all the causes of $Z_i$, $N_i$ is the corresponding latent noise variable. The causal structure among $Z_i$ can be represented by a Directed Acyclic Graph (DAG). The structure of the DAG is unknown. Our target is modeling the latent generative process of the instances and noisy labels.

**Generative Process** In our approach, the generative process of noisy data involves the latent clean labels $Y$, a set of latent factors $\boldsymbol{Z}$, the variable of instances $\boldsymbol{X}$, and the noisy labels $\tilde{Y}$. Our proposed generative process is flexible. In such a way, it allows different classes to have different class conditional distributions $p(\boldsymbol{Z}|Y)$. For instance, the distribution of latent factors for "house" and "plane" can be markedly different. Moreover, we also allow to have causal associations among latent causal factors. This usually happens in many real-world cases. As an illustration, a factor related to the sun might influence a factor about brightness. Instances and noisy labels are then generated from these causal factors via two distinct nonlinear mixing functions.

We also allow the causal factors used to generate instances and noisy labels to not be shared. Considering the real-world case, the factors used to generate instances and the factors used to generate noisy labels are different. The overall generative process is nonlinear. Specifically, the generative processes of instances $\boldsymbol{X}$ and noisy labels $\tilde{Y}$ are nonlinear. But the generative process of causal factors $\boldsymbol{Z}$ is linear, which can provide an identifiable guarantee. The generative process of noisy data is shown in Fig. 2. The black arrow indicates the causal direction in the structural causal model, while the blue arrow indicates the weights of the edge are changed across the clean label $Y$. The observed instances $\boldsymbol{X}$ and the noisy label $\tilde{Y}$ are generated by causal factors $\boldsymbol{Z}$. The causal factor $Z_i$ is generated by its parent and corresponding noise variable $N_i$. Since causal factor $\boldsymbol{Z}$ is latent, discovering the causal structure is a challenging problem. To provide a theoretical guarantee, we assume the causal mechanism among causal factors is linear. This generative process of the causal factor $\boldsymbol{Z}$ can be modeled by a linear Structure Causal Model (SCM):

$$Z_i := \boldsymbol{W}_i^T \boldsymbol{Z} + N_i, \tag{1}$$

where $\boldsymbol{W}$ is a matrix used to represent the association among causal factors. The value of $W_{i,j}$ represents the weight of causal association between the causal factor $Z_i$ and its parent $Z_j$. We assume the causal structure among causal factors is a fully-connected DAG, then $\boldsymbol{W}$ is an upper triangular matrix with zero-value diagonal elements. The diagonal elements are zero-value because the parents of a causal factor do not contain itself.

The clean label $Y$ modulate the weight matrix $\boldsymbol{W}$ via a function $\boldsymbol{f_W}(\cdot)$, *i.e.*, $\boldsymbol{W} = \boldsymbol{f_W}(Y)$. The function $\boldsymbol{f_W}(\cdot)$ is called weight model in our paper. Then, the prior distribution of causal factors $p_{\boldsymbol{f_W},\boldsymbol{\beta}}(\boldsymbol{Z}|Y)$ can be modeled, where $\boldsymbol{\beta}$ is the parameter of the distribution of latent noise variable $\boldsymbol{N}$ and $\boldsymbol{\beta}$ is modulated by clean label.

**Intuition about Inferring Latent Generative Process** Emerging theories in latent causality (Yang et al., 2021; Liu et al., 2022b) suggest that we can efficiently infer the latent generative process with the aid of additional supervised information. In the context of learning with noisy labels, this additional supervised information necessitates some clean examples. Fortunately, the task of selecting clean examples from noisy training data is well-established and well-studied. Many current techniques

can effectively select a small set of examples with a high clean ratio. By having additional supervised information derived from selected clean examples, inferring the generative process becomes feasible.

There we provide some intuition about the core idea of existing identifiability results (Liu et al., 2022b) that why the generative process can be effectively inferred. At its heart is the realization that the parameters governing this generative process are not unique to individual examples but are shared across them. In our case, the same class has the same generative process Intuitively, the essence of inferring the generative process lies in inferring these shared parameters. When these parameters are consistent across different examples, their count remains unchanged, and the complexity of the generative process remains constant. With a subset of selected clean examples along with their noisy labels, one can calculate the most probable parameter value for generating these examples. This likelihood narrows down the uncertainty of the parameter value. In Appendix C, we discuss the specific assumption to make the latent generative process of noisy data fully identifiable.

We propose a practical method to learn the latent Generative Process (GenP) of noisy data. The working flow of our method is shown in Fig. 3. We use a trained classification network to estimate the clean labels. We use a network as the encoder to infer the causal factors for the instances. An instances decoder is used to map the causal factors to instances. A noisy label decoder is used to map the causal factors to noisy labels. These networks can be learned through the Variational AutoEncoder framework. After training, the encoder and decoders serve as regularizers for the classification network tasked with predicting clean labels. The underlying intuition is as follows: given an instance $\boldsymbol{x}$, the classification network outputs an estimated clean label $y$, then we utilize the encoder to derive its corresponding casual factors $\boldsymbol{z}$ based on the instance and inferred clean label. The inferred casual factors $\boldsymbol{z}$ are fed into the decoders and produce the most probable instance and noisy label. By minimizing the difference between the predictions and the ground truths, the encoder and decoders effectively guide the neural network to predict a clean label that aligns closely with the generative process of the instance and noisy label.

## 3.1 CLEAN EXAMPLES SELECTION

With only the observed variables, recovering the latent variables and the generative process is ill-posed (Hyvärinen & Pajunen, 1999; Locatello et al., 2019). Recent advances in causal representation learning (Yang et al., 2021; Liu et al., 2022b) demonstrate the latent variables and the generative process with the help of additional supervision information. In the learning with noisy labels setting, the clean label $Y$ can act as additional supervision information, but the clean label is unknown. We have a noisy dataset $\tilde{\mathcal{D}} = \{(\boldsymbol{x}^{(i)}, \tilde{y}^{(i)})\}_{i=1}^{n}$, where $n$ is the number of examples. Let $q_{\tilde{\mathcal{D}}}(\boldsymbol{X}, \tilde{Y})$ denotes the empirical noisy data distribution given by the noisy dataset $\tilde{\mathcal{D}}$. We use the small-loss trick to select clean examples to obtain additional supervision information. Specifically, the classification network trained on noisy data will first memorize the examples with correct labels and then gradually memorize the incorrect ones. Thus, the losses for examples with correct labels are probably smaller than those with correct labels. The clean examples can be distinguished from noisy examples by leveraging the loss. By employing the labels in the selected clean examples, the classification network $\hat{q}_{\psi}(\cdot)$ could be obtained.

To improve the performance of the classification network, the information of the remaining examples is exploited to train the classification network by using the semi-supervised learning method Mix-Match (Berthelot et al., 2019). Specifically, Let the selected clean examples as the labeled examples $\mathcal{S}_X$ and the remaining examples as the unlabeled examples $\mathcal{S}_U$. The labels in the labeled examples $\mathcal{S}_X$ are refined through the output of the classification network $\hat{q}_{\psi}(\cdot)$. The outputs of the classification network $\hat{q}_{\psi}(\cdot)$ for unlabeled examples are used to generate guessed labels. Then, the temperature sharpening is applied to the refined labels and guessed labels on the labeled examples and unlabeled examples, respectively. After that, the labeled examples $\mathcal{S}_X$ and the unlabeled examples $\mathcal{S}_U$ are transformed into augmented labeled examples $\mathcal{S}'_X$ and augmented unlabeled examples $\mathcal{S}'_U$ by using a linear mixing. The loss function used to train the classification network is

$$\mathcal{L}_{semi} = \mathcal{L}_{\mathcal{S}_X} + \lambda_u \mathcal{L}_{\mathcal{S}_U} + \lambda_r \mathcal{L}_{\text{reg}}, \tag{2}$$

where $\mathcal{L}_{\mathcal{S}_X}$ is the cross-entropy loss for the labeled examples; $\mathcal{L}_{\mathcal{S}_U}$ is the mean squared error the unlabeled examples; $\mathcal{L}_{\text{reg}}$ is a regularization term to prevent the model from predicting all examples

to belong to a single class. These three terms are defined as follows specifically.

$$\mathcal{L}_{\mathcal{S}_X} = -\frac{1}{|\mathcal{S}'_X|} \sum_{\boldsymbol{x},\boldsymbol{p}\in\mathcal{S}'_X} \sum_i p_i \log(q_\psi(Y=i|\boldsymbol{x})), \tag{3}$$

$$\mathcal{L}_{\mathcal{S}_U} = \frac{1}{|\mathcal{S}'_U|} \sum_{\boldsymbol{x},\boldsymbol{p}\in\mathcal{S}'_U} \|\boldsymbol{p} - q_\psi(Y|\boldsymbol{x})\|_2^2, \tag{4}$$

$$\mathcal{L}_{\text{reg}} = \sum_i \frac{1}{C} \log(1 \Big/ \frac{C}{|\mathcal{S}'_X| + |\mathcal{S}'_U|} \sum_{\boldsymbol{x}\in\mathcal{S}_X+\mathcal{S}_U} q_\psi(Y=i|\boldsymbol{x})), \tag{5}$$

where $\boldsymbol{p}$ is the label, $q_\psi(Y|\boldsymbol{x}) := [q_\psi(Y=1|\boldsymbol{x}),\dots,q_\psi(Y=C|\boldsymbol{x})]^T$, and $C$ denote number of class.

## 3.2 DATA GENERATIVE PROCESS MODELING

To model the latent causal factors and their causal relation, we use an encoder network to infer the latent causal factors. Then, we use a network to infer the weight of causal associations among the causal factors. After that, we could build a Causal Structure Model (SCM) for causal factors by employing the inferred latent factors with the weight of causal associations.

We first introduce the encoder $\hat{q}_\varphi(\cdot)$. The encoder takes an instance $\tilde{x}$ and corresponding clean labels $y$ as inputs, then output the distribution of causal factors $\boldsymbol{z}$, $i.e.$, $q_\varphi(\boldsymbol{Z}=\boldsymbol{z}|\boldsymbol{X}=\boldsymbol{x},Y=y)$. Thus, the encoder can be used to model the distribution $q_\varphi(\boldsymbol{Z}|\boldsymbol{X},Y)$. The weight model $\boldsymbol{f_W}$ is learned to infer the matrix $\boldsymbol{W}$.

**Modeling Generation of Observed Variables**   We assume that the instances and noisy labels have different and independent generative processes. The instances and noisy labels are generated through different causal factors. Each causal factor at least generates an observed variable, $i.e.$, the observed instance or the observed noisy label. The process of selecting different causal factors can be implemented by a mask operation. Let $\boldsymbol{M_X}$ denote the mask for selecting causal factors to generate the instances, and $\boldsymbol{M_{\tilde{Y}}}$ denotes the mask for selecting causal factors to generate the noisy labels. The masking process can be represented as follows.

$$\boldsymbol{Z_X} = \boldsymbol{M_X} \odot \boldsymbol{Z}, \boldsymbol{Z_{\tilde{Y}}} = \boldsymbol{M_{\tilde{Y}}} \odot \boldsymbol{Z}, \tag{6}$$

where $\odot$ is the element-wise multiplication. To ensure the sparsity of the masks, we utilize L1 loss on the masks. The sparsity is used as we consider the causal factors for generating $\tilde{Y}$ and $\boldsymbol{X}$ can be different. Additionally, it encourages learning simple generative processes that have been commonly used in casualty literature.

The generative process of instance $\boldsymbol{x}$ and noisy label $\tilde{y}$ is shown in the following.

$$\boldsymbol{x} \sim p_{\boldsymbol{f_X}}(\boldsymbol{X}|\boldsymbol{Z}=\boldsymbol{z_X}), \tilde{y} \sim p_{\boldsymbol{f_{\tilde{Y}}}}(\tilde{Y}|\boldsymbol{Z}=\boldsymbol{z_{\tilde{Y}}}). \tag{7}$$

The generative probabilistic model for instances and noisy labels can be expressed as:

$$p_{\boldsymbol{f}}(\boldsymbol{X},\tilde{Y}|\boldsymbol{Z}) = p_{\boldsymbol{f_X}}(\boldsymbol{X}|\boldsymbol{M_X}\odot\boldsymbol{Z})p_{\boldsymbol{f_{\tilde{Y}}}}(\tilde{Y}|\boldsymbol{M_{\tilde{Y}}}\odot\boldsymbol{Z}) \tag{8}$$

To model these distributions, we utilize two decoder networks: an instance decoder, $\hat{p}_{\boldsymbol{f_X}}(\cdot)$, and a noisy label decoder, $\hat{p}_{\boldsymbol{f_{\tilde{Y}}}}(\cdot)$. The former outputs the instance $\boldsymbol{x}$ based on the causal factors $\boldsymbol{z}$, while the latter outputs the noisy label $\tilde{Y}$ using the same causal factors.

The overall generative model is a probabilistic model parameterized by $\theta = (\boldsymbol{f}, \boldsymbol{f_W}, \boldsymbol{\beta})$:

$$p_\theta(\boldsymbol{X},\tilde{Y},\boldsymbol{Z}|Y) = p_{\boldsymbol{f}}(\boldsymbol{X},Y|\boldsymbol{Z})p_{\boldsymbol{f_W},\boldsymbol{\beta}}(\boldsymbol{Z}|Y) \tag{9}$$

**Optimization for Generative Process**   The encoder, instance decoder, noisy label decoder, and weight model have to be trained on the dataset with clean labels. We denote this dataset as $\mathcal{D} = \{(\boldsymbol{x}^{(i)}, \tilde{y}^{(i)}, y^{(i)})\}_{i=1}^n$. Let $q_{\mathcal{D}}(\boldsymbol{X},\tilde{Y},Y)$ denotes the empirical data distribution given by the dataset $\mathcal{D}$. However, the clean label $Y$ is unknown in the learning with noisy labels setting. We only have a

Table 1: Means and standard deviations (percentage) of classification accuracy on Fashion-MNIST.

| | Fashion-MNIST | | | | |
| | IDN-10% | IDN-20% | IDN-30% | IDN-40% | IDN-50% |
|---|---|---|---|---|---|
| CE | $93.16 \pm 0.02$ | $92.68 \pm 0.16$ | $91.41 \pm 0.16$ | $87.60 \pm 0.33$ | $71.76 \pm 0.77$ |
| MentorNet | $93.16 \pm 0.01$ | $91.57 \pm 0.29$ | $90.52 \pm 0.41$ | $88.14 \pm 0.76$ | $61.62 \pm 1.42$ |
| CoTeaching | $94.26 \pm 0.06$ | $91.21 \pm 0.31$ | $90.30 \pm 0.42$ | $89.10 \pm 0.29$ | $63.22 \pm 1.56$ |
| Reweight | $93.42 \pm 0.16$ | $93.12 \pm 0.18$ | $92.19 \pm 0.18$ | $88.51 \pm 1.52$ | $75.00 \pm 5.28$ |
| Forward | $93.48 \pm 0.11$ | $92.82 \pm 0.12$ | $91.05 \pm 1.44$ | $87.82 \pm 1.81$ | $78.34 \pm 2.98$ |
| PTD | $92.01 \pm 0.35$ | $91.08 \pm 0.46$ | $89.66 \pm 0.43$ | $85.69 \pm 0.77$ | $75.96 \pm 1.38$ |
| CausalNL | $91.63 \pm 0.18$ | $90.84 \pm 0.31$ | $90.68 \pm 0.37$ | $90.01 \pm 0.45$ | $78.19 \pm 1.01$ |
| CCR | $88.48 \pm 0.16$ | $83.59 \pm 0.25$ | $75.40 \pm 0.19$ | $64.39 \pm 0.30$ | $50.17 \pm 0.29$ |
| MEIDTM | $86.00 \pm 0.84$ | $80.99 \pm 0.64$ | $73.12 \pm 2.34$ | $63.81 \pm 3.02$ | $58.60 \pm 3.32$ |
| BLTM | $91.28 \pm 1.93$ | $91.20 \pm 0.27$ | $85.51 \pm 4.77$ | $82.42 \pm 1.51$ | $67.65 \pm 5.65$ |
| DivideMix | $95.04 \pm 0.09$ | $94.85 \pm 0.15$ | $94.22 \pm 0.14$ | $92.28 \pm 0.13$ | $85.76 \pm 0.31$ |
| GenP | $\mathbf{95.32 \pm 0.11}$ | $\mathbf{95.14 \pm 0.10}$ | $\mathbf{94.66 \pm 0.12}$ | $\mathbf{93.78 \pm 0.14}$ | $\mathbf{88.97 \pm 0.28}$ |

Table 2: Means and standard deviations (percentage) of classification accuracy on CIFAR-10.

| | CIFAR-10 | | | | |
| | IDN-10% | IDN-20% | IDN-30% | IDN-40% | IDN-50% |
|---|---|---|---|---|---|
| CE | $87.81 \pm 0.15$ | $85.90 \pm 0.30$ | $82.67 \pm 0.31$ | $74.49 \pm 0.95$ | $46.81 \pm 2.52$ |
| MentorNet | $86.87 \pm 0.14$ | $83.89 \pm 0.16$ | $77.83 \pm 0.28$ | $61.96 \pm 0.97$ | $47.89 \pm 2.03$ |
| CoTeaching | $90.06 \pm 0.32$ | $87.16 \pm 0.50$ | $81.80 \pm 0.26$ | $63.95 \pm 2.87$ | $45.92 \pm 2.21$ |
| Reweight | $89.63 \pm 0.27$ | $87.85 \pm 0.97$ | $81.29 \pm 6.49$ | $80.33 \pm 3.75$ | $75.14 \pm 2.40$ |
| Forward | $88.89 \pm 0.18$ | $87.83 \pm 0.30$ | $82.01 \pm 3.29$ | $79.49 \pm 1.85$ | $71.11 \pm 8.78$ |
| PTD | $79.01 \pm 0.20$ | $76.05 \pm 0.53$ | $72.28 \pm 0.49$ | $58.62 \pm 0.88$ | $53.98 \pm 2.34$ |
| CausalNL | $83.39 \pm 0.34$ | $80.91 \pm 1.14$ | $79.05 \pm 0.54$ | $79.08 \pm 0.50$ | $76.56 \pm 0.02$ |
| CCR | $91.43 \pm 0.05$ | $90.93 \pm 0.07$ | $90.15 \pm 0.11$ | $89.01 \pm 0.15$ | $86.05 \pm 0.18$ |
| MEIDTM | $86.52 \pm 0.38$ | $82.93 \pm 0.44$ | $77.35 \pm 0.21$ | $68.21 \pm 2.09$ | $57.84 \pm 3.51$ |
| BLTM | $80.16 \pm 0.37$ | $77.50 \pm 1.30$ | $71.47 \pm 2.33$ | $63.20 \pm 4.52$ | $48.12 \pm 9.03$ |
| DivideMix | $96.03 \pm 0.14$ | $95.92 \pm 0.12$ | $95.66 \pm 0.15$ | $95.03 \pm 0.12$ | $86.98 \pm 0.28$ |
| GenP | $\mathbf{96.12 \pm 0.12}$ | $\mathbf{96.05 \pm 0.12}$ | $\mathbf{95.74 \pm 0.13}$ | $\mathbf{95.44 \pm 0.12}$ | $\mathbf{89.39 \pm 0.45}$ |

noisy dataset $\tilde{\mathcal{D}} = \{(\boldsymbol{x}^{(i)}, \tilde{y}^{(i)})\}_{i=1}^n$. Therefore, we use the learned classification network $q_\psi(Y|\boldsymbol{X})$ to predict the clean label for each instance. The distribution $q_{\mathcal{D}}(\boldsymbol{X}, \tilde{Y}, Y)$ can be approximate through the noisy data distribution $q_{\tilde{\mathcal{D}}}(\boldsymbol{X}, \tilde{Y})$ and distribution $q_\psi(Y|\boldsymbol{X})$ modeled by the classification network:

$$q_{\mathcal{D}}(\boldsymbol{X}, \tilde{Y}, Y) = q_{\tilde{\mathcal{D}}}(\boldsymbol{X}, \tilde{Y})q_{\mathcal{D}}(Y|\boldsymbol{X}, \tilde{Y}) \approx q_{\tilde{\mathcal{D}}}(\boldsymbol{X}, \tilde{Y})q_\psi(Y|\boldsymbol{X}). \tag{10}$$

All the networks can be learned by maximizing the following Evidence Lower BOund (ELBO):

$$\mathbb{E}_{(\boldsymbol{x},\tilde{y},y)\sim q_{\mathcal{D}}}[p_\theta(\boldsymbol{x}, \tilde{y}|y)] \geq ELBO = \mathbb{E}_{(\boldsymbol{x},\tilde{y},y)\sim q_{\mathcal{D}}}\left[\mathbb{E}_{\boldsymbol{z}\sim q_\varphi}[\log p_{\boldsymbol{f}}(\boldsymbol{x}, \tilde{y}|\boldsymbol{z})]\right.$$
$$-KL(q_\varphi(\boldsymbol{z}|\boldsymbol{x}, y)||p_{\boldsymbol{f_W},\boldsymbol{\beta}}(\boldsymbol{z}|y))]$$
$$\approx \mathbb{E}_{(\boldsymbol{x},\tilde{y},y)\sim q_{\tilde{\mathcal{D}}}q_\psi}\left[\mathbb{E}_{\boldsymbol{z}\sim q_\varphi}[\log p_{\boldsymbol{f}}(\boldsymbol{x}, \tilde{y}|\boldsymbol{z})] - KL(q_\varphi(\boldsymbol{z}|\boldsymbol{x}, y)||p_{\boldsymbol{f_W},\boldsymbol{\beta}}(\boldsymbol{z}|y))\right],$$

where $KL$ denotes the Kullback–Leibler divergence.

**Optimization in End-to-End Manner** The working flow of our method is shown in Fig. 3. We optimize our model end-to-end rather than learning the classification network and the generation process alternately. The final loss function used to train the networks is

$$\mathcal{L} = \mathcal{L}_{semi} - \lambda_{ELBO}ELBO + \lambda_M(\|\boldsymbol{M_X}\|_1 + \|\boldsymbol{M_{\tilde{Y}}}\|_1), \tag{11}$$

where $\lambda_{ELBO}$ and $\lambda_M$ are hyperparameters. In our experiments, $\lambda_{ELBO}$ and $\lambda_M$ is set to 0.01.

## 4 EXPERIMENTS

In this section, we compare the classification performance of the proposed method with that of state-of-the-art methods on synthetic and real-world noisy datasets. Due to space limitation, the optimization details are in the Appendix B.

Table 3: Means and standard deviations (percentage) of classification accuracy on and CIFAR-100.

| | CIFAR-100 | | | | |
|---|---|---|---|---|---|
| | IDN-10% | IDN-20% | IDN-30% | IDN-40% | IDN-50% |
| CE | $57.43 \pm 0.38$ | $54.98 \pm 0.19$ | $50.65 \pm 0.25$ | $43.65 \pm 0.15$ | $34.91 \pm 0.16$ |
| MentorNet | $58.45 \pm 0.06$ | $55.98 \pm 0.32$ | $51.40 \pm 0.68$ | $43.79 \pm 0.48$ | $34.06 \pm 0.52$ |
| CoTeaching | $65.22 \pm 0.41$ | $62.36 \pm 0.38$ | $57.02 \pm 0.43$ | $49.84 \pm 0.77$ | $38.28 \pm 1.08$ |
| Reweight | $59.38 \pm 0.33$ | $55.14 \pm 0.07$ | $46.91 \pm 1.26$ | $37.80 \pm 1.01$ | $28.45 \pm 2.57$ |
| Forward | $59.58 \pm 0.42$ | $56.59 \pm 0.25$ | $52.75 \pm 0.25$ | $46.03 \pm 0.65$ | $35.07 \pm 0.91$ |
| PTD | $67.33 \pm 0.33$ | $65.33 \pm 0.59$ | $64.56 \pm 1.55$ | $59.73 \pm 0.76$ | $56.80 \pm 1.32$ |
| CausalNL | $47.29 \pm 1.11$ | $41.47 \pm 0.43$ | $40.98 \pm 0.62$ | $34.02 \pm 0.95$ | $32.13 \pm 2.23$ |
| CCR | $69.73 \pm 0.07$ | $68.84 \pm 0.09$ | $67.65 \pm 0.08$ | $66.54 \pm 0.09$ | $64.66 \pm 0.11$ |
| MEIDTM | $69.88 \pm 0.45$ | $69.16 \pm 0.16$ | $66.76 \pm 0.30$ | $63.46 \pm 0.48$ | $59.18 \pm 0.16$ |
| BLTM | $48.82 \pm 0.44$ | $46.61 \pm 1.10$ | $41.35 \pm 0.85$ | $35.67 \pm 1.97$ | $29.28 \pm 0.74$ |
| DivideMix | $77.15 \pm 0.29$ | $76.73 \pm 0.24$ | $76.13 \pm 0.22$ | $72.10 \pm 0.33$ | $61.10 \pm 0.35$ |
| GenP | $\mathbf{78.80 \pm 0.20}$ | $\mathbf{77.98 \pm 0.20}$ | $\mathbf{77.71 \pm 0.23}$ | $\mathbf{75.05 \pm 0.24}$ | $\mathbf{65.68 \pm 0.39}$ |

Table 4: Means and standard deviations (percentage) of classification accuracy on CIFAR-10N.

| | CIFAR-10N | | | | |
|---|---|---|---|---|---|
| | Worst | Aggregate | Random 1 | Random 2 | Random 3 |
| CE | $79.39 \pm 0.35$ | $87.91 \pm 0.18$ | $86.05 \pm 0.13$ | $86.12 \pm 0.12$ | $86.12 \pm 0.16$ |
| MentorNet | $77.91 \pm 0.38$ | $75.56 \pm 0.25$ | $77.10 \pm 0.25$ | $77.06 \pm 0.13$ | $77.06 \pm 0.13$ |
| CoTeaching | $81.86 \pm 0.40$ | $82.45 \pm 0.08$ | $82.90 \pm 0.46$ | $82.95 \pm 0.26$ | $82.66 \pm 0.12$ |
| Reweight | $77.68 \pm 2.46$ | $89.34 \pm 0.09$ | $88.44 \pm 0.10$ | $88.16 \pm 0.10$ | $88.03 \pm 0.10$ |
| Forward | $79.27 \pm 1.18$ | $89.22 \pm 0.21$ | $86.84 \pm 0.97$ | $86.99 \pm 0.10$ | $87.53 \pm 0.34$ |
| PTD | $65.62 \pm 5.28$ | $84.66 \pm 3.28$ | $82.11 \pm 3.17$ | $74.76 \pm 9.98$ | $84.29 \pm 0.64$ |
| CausalNL | $72.09 \pm 0.84$ | $82.20 \pm 0.32$ | $81.10 \pm 0.09$ | $81.13 \pm 0.10$ | $81.03 \pm 0.41$ |
| CCR | $80.43 \pm 0.24$ | $90.10 \pm 0.09$ | $88.53 \pm 0.08$ | $88.21 \pm 0.11$ | $88.46 \pm 0.08$ |
| MEIDTM | $79.59 \pm 0.89$ | $90.15 \pm 0.27$ | $87.81 \pm 0.52$ | $88.07 \pm 0.18$ | $87.86 \pm 0.21$ |
| BLTM | $68.21 \pm 1.67$ | $79.41 \pm 1.00$ | $78.09 \pm 1.03$ | $76.99 \pm 1.23$ | $76.26 \pm 0.71$ |
| DivideMix | $93.41 \pm 0.19$ | $95.12 \pm 0.15$ | $95.32 \pm 0.13$ | $95.15 \pm 0.09$ | $95.23 \pm 0.16$ |
| GenP | $\mathbf{93.87 \pm 0.13}$ | $\mathbf{95.39 \pm 0.18}$ | $\mathbf{95.38 \pm 0.13}$ | $\mathbf{95.30 \pm 0.12}$ | $\mathbf{95.26 \pm 0.13}$ |

Table 5: Means and standard deviations (percentage) of classification accuracy on Clothing1M.

| CE | Decoupling | MentorNet | Co-teaching | Forward |
|---|---|---|---|---|
| 68.88 | 54.53 | 56.79 | 60.15 | 69.91 |
| T-Revision | BLTM | CausalNL | DivideMix | GenP |
| 70.97 | 73.39 | 72.24 | 74.76 | **74.81** |

## 4.1 EXPERIMENT SETUP

**Datasets.** We empirically verify the performance of our method on three synthesis datasets, i.e., Fashion-MNIST (Xiao et al., 2017), CIFAR-10 (Krizhevsky et al., 2009), CIFAR-100 (Krizhevsky et al., 2009), and two real-world datasets, i.e., CIFAR-10N (Wei et al., 2022) and Clothing1M (Xiao et al., 2015). Fashion-MNIST contains 70,000 28x28 grayscale images with 10 classes total, 60,000 images for training, and 10,000 images for testing. Both CIFAR-10 and CIFAR-100 contain 50,000 training images and 10,000 testing images. The image size is 32x32. CIFAR-10 has 10 classes of images, and CIFAR-100 has 100 classes of images. The three datasets contain clean labels. We corrupted the training data manually according to the instance-dependent noisy label generation method proposed in Xia et al. (2020). All experiments are repeated five times. CIFAR-10N is a real-world label-noise version of CIFAR-10. It contains human-annotated noisy labels with five different types of noise (Worst, Aggregate, Random 1, Random 2, and Random 3). The corresponding noise rates are 40.21%, 9.03%, 17.23%, 18.12%, and 17.64%. Clothing1M contains 1 million images with real-world noisy labels, including 50,000, 14,000, and 10,000 images with clean labels for training, validation, and testing, respectively. We assume that the clean data is unavailable, and therefore, we do not use the clean data for training and validation.

**Baselines.** The baselines used in our experiments for comparison are: 1). CE, training the classification network using standard cross-entropy loss on noisy data directly; 2), MentorNet (Jiang et al., 2018), pretraining a classification network to select reliable examples for the main classification network; 3), Co-teaching (Han et al., 2018), which uses two classification networks to select reliable examples for each other; 4), Reweight (Liu & Tao, 2016), estimating a unbiased risk defined on clean data using noisy data by using importance reweighting method; 5), Forward (Patrini et al., 2017), which assumes the noise transition is class-dependent, the correct loss function; 6), PTD (Xia et al., 2020), estimating instance-dependent noisy transition through the parts of instances; 7) CausalNL Yao et al. (2021), which explores the information in the instances to help the learning of classification network; 8), CCR (Cheng et al., 2022b) uses forward-backward cycle-consistency regularization to learn noise transition; 9), MEIDTM (Cheng et al., 2022a), which uses Lipschitz continuity to constrain the noise transition in the same manifold to be the same; 10), BLTM (Yang et al., 2022), which learn the noise transition on a part of dataset with Bayes optimal label; 11), DivideMix (Li et al., 2020), which divides the noisy examples into labeled examples and unlabeled examples, and train the classification network using semi-supervised technique MixMatch (Berthelot et al., 2019).

**Implementation.** We implement our algorithm using PyTorch and conduct all our experiments on RTX 4090. We use a PreAct ResNet-18 (He et al., 2016b) as the classification network for Fashion-MNIST, CIFAR-10, CIFAR-100, and CIFAR-10N, a ResNet-50 (He et al., 2016a) with ImageNet pre-trained weight as the classification network for Clothing1M. We use a 4-hidden-layer convolutional network as the encoder, and the channel sizes of corresponding feature maps are 32, 64, 128, and 256 for Fashion-MNIST, CIFAR-10, CIFAR-100, and CIFAR-10N. We use a 5-hidden-layer convolutional network as the encoder, and the channel sizes of corresponding feature maps are 32, 64, 128, 256, and 512 for Clothing1M. A 4-hidden-layer transposed-convolutional network as the instance decoder and the channel size of corresponding feature maps are 256, 128, 64, and 32 for Fashion-MNIST, CIFAR-10, CIFAR-100, and CIFAR-10N. A 5-hidden-layer transposed-convolutional network as the instance decoder and the channel size of corresponding feature maps are 512, 256, 128, 64, and 32 for Clothing1M. We use a three-layer MLP with the Leak ReLU activation function as the weight model to infer the weight of causal associations among the causal factors. To infer the noisy label, a three-layer MLP with the Leak ReLU activation function is used as the noisy label decoder. The number of causal factors is set to 4 in all our experiments.

## 4.2 CLASSIFICATION ACCURACY

We conducted extensive experiments on three synthetic noise datasets, *i.e.*, Fashion-MNIST, CIFAR-10, and CIFAR-100, and two real-world datasets, *i.e.*, CIFAR-10N and Clothing1M. We employed instance-dependent noisy label generation methods for the synthetic datasets, as proposed by (Xia et al., 2020). We experimented with noise rates of 0.1, 0.2, 0.3, 0.4, and 0.5, denoted by IDN-0.1, IDN-0.2, IDN-0.3, IDN-0.4, and IDN-0.5 respectively. The experimental results for synthetic datasets are presented in Tab. 1, Tab. 2 and Tab. 3. The real-world dataset experiment results are in Tab. 4 and Tab. 5. Our proposed method outperforms existing methods in terms of test accuracy on both synthetic and real-world datasets containing label noise. The experiment results demonstrate that the proposed method can capture the noise transition under different settings and improve the performance of the classification network.

## 5 CONCLUSION

Noise transition is important for many label-noise learning algorithms. However, current label-noise learning methods often can only estimate the noise transitions for some instances. It is crucial to understand the connection among the noise transitions for different instances to apply these estimated noise transitions to other instances. Prior work tackled this issue by introducing new assumptions to define the similarity of noise transitions across different instances. However, whether these predefined similarities are truthfulness to the underlying connection is unclear. Given only noisy data, the introduced assumptions are hard to verify. If similarities are not truthful, the estimation error of the noise transition could be large, leading to performance degeneration for label-noise learning algorithms. We propose a novel method to build the connection among the noise transitions. The connection is built by modeling the causal generative process of noisy data. Experiments on both synthesis and real-world datasets demonstrate the effectiveness of our method.

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

## A  DERIVATION OF ELBO

The derivation of ELBO is shown as follows:

$$
\begin{aligned}
\mathbb{E}_{(\boldsymbol{x},\tilde{y},y)\sim q_{\mathcal{D}}}[\log p_\theta(\boldsymbol{x},\tilde{y}|y)] &= \mathbb{E}_{(\boldsymbol{x},\tilde{y},y)\sim q_{\mathcal{D}}}\left[\log\frac{p_\theta(\boldsymbol{x},\tilde{y},\boldsymbol{z}|y)}{q_\varphi(\boldsymbol{z}|\boldsymbol{x},y)}\frac{q_\phi(\boldsymbol{z}|\boldsymbol{x},y)}{p_\theta(\boldsymbol{z}|\boldsymbol{x},\tilde{y},y)}\right]\\
&= \mathbb{E}_{(\boldsymbol{x},\tilde{y},y)\sim q_{\mathcal{D}}}\left[\int q_\varphi(\boldsymbol{z}|\boldsymbol{x},y)\log\frac{p_\theta(\boldsymbol{x},\tilde{y},\boldsymbol{z}|y)}{q_\varphi(\boldsymbol{z}|\boldsymbol{x},y)}\frac{q_\varphi(\boldsymbol{z}|\boldsymbol{x},y)}{p_\theta(\boldsymbol{z}|\boldsymbol{x},\tilde{y},y)}d\boldsymbol{z}\right]\\
&= \mathbb{E}_{(\boldsymbol{x},\tilde{y},y)\sim q_{\mathcal{D}}}\left[\int q_\varphi(z|x,y)\log\frac{p_\theta(\boldsymbol{x},\tilde{y},\boldsymbol{z}|y)}{q_\varphi(\boldsymbol{z}|\boldsymbol{x},y)}d\boldsymbol{z}\right]\\
&\quad +KL(q_\varphi(\boldsymbol{z}|\boldsymbol{x},y)||p_\theta(\boldsymbol{z}|\boldsymbol{x},\tilde{y},y))]\\
&\geq \mathbb{E}_{(\boldsymbol{x},\tilde{y},y)\sim q_{\mathcal{D}}}\left[\int q_\varphi(\boldsymbol{z}|\boldsymbol{x},y)\log\frac{p_\theta(\boldsymbol{x},\tilde{y},\boldsymbol{z}|y)}{q_\varphi(\boldsymbol{z}|\boldsymbol{x},y)}d\boldsymbol{z}\right].
\end{aligned}
$$

$$
\begin{aligned}
ELBO &= \mathbb{E}_{(\boldsymbol{x},\tilde{y},y)\sim q_{\mathcal{D}}}\left[\int q_\varphi(\boldsymbol{z}|\boldsymbol{x},y)\log\frac{p_\theta(\boldsymbol{x},\tilde{y},\boldsymbol{z}|y)}{q_\varphi(\boldsymbol{z}|\boldsymbol{x},y)}d\boldsymbol{z}\right]\\
&= \mathbb{E}_{(\boldsymbol{x},\tilde{y},y)\sim q_{\mathcal{D}}}\left[\int q_\varphi(\boldsymbol{z}|\boldsymbol{x},y)\log\frac{p_{\boldsymbol{f}}(\boldsymbol{x},\tilde{y}|\boldsymbol{z},y)p_{\boldsymbol{f_W},\boldsymbol{\beta}}(\boldsymbol{z}|y)}{q_\varphi(\boldsymbol{z}|\boldsymbol{x},y)}d\boldsymbol{z}\right]\\
&= \mathbb{E}_{(\boldsymbol{x},\tilde{y},y)\sim q_{\mathcal{D}}}\left[\int q_\varphi(\boldsymbol{z}|\boldsymbol{x},y)\log p_{\boldsymbol{f}}(\boldsymbol{x},\tilde{y}|\boldsymbol{z},y)d\boldsymbol{z}+\int q_\varphi(\boldsymbol{z}|x,y)\frac{p_{\boldsymbol{f_W},\boldsymbol{\beta}}(\boldsymbol{z}|y)}{q_\varphi(\boldsymbol{z}|x,y)}d\boldsymbol{z}\right]\\
&= \mathbb{E}_{(\boldsymbol{x},\tilde{y},y)\sim q_{\mathcal{D}}}\left[\mathbb{E}_{\boldsymbol{z}\sim q_\varphi}[\log p_{\boldsymbol{f}}(\boldsymbol{x},\tilde{y}|\boldsymbol{z},y)]-KL(q_\varphi(\boldsymbol{z}|\boldsymbol{x},y)||p_{\boldsymbol{f_W},\boldsymbol{\beta}}(\boldsymbol{z}|y))\right]
\end{aligned}
$$

## B  IMPLEMENTATION DETAILS

**Distributions modeling**  We provide the distribution modeling details here, including the distribution of latent noise factors, the prior distribution of causal factors, and the inferred posterior distribution of causal factors.

We assume the distribution of $N_i$ is Gaussian, which also is modulated by the auxiliary variable $Y$.

$$
N_i \sim \mathcal{N}(\beta_{i,1}(Y), \beta_{i,2}(Y)),  \tag{12}
$$

where $\beta_{i,1}$ and $\beta_{i,2}$ are the mean and variance of noise term $N_i$, respectively.

Since the distribution of noise variable $N_i$ is Gaussian distribution modulated by the auxiliary variable $Y$, the causal weights are also modulated by the auxiliary variable $Y$. Then, the conditional probability distribution $p(\boldsymbol{Z}|Y)$ is a multivariate Gaussian distribution:

$$
p_{\boldsymbol{f_W},\boldsymbol{\beta}}(\boldsymbol{Z}|Y) = \mathcal{N}(\boldsymbol{\mu}, \Sigma),  \tag{13}
$$

where $\mu$ is the mean, $\Sigma$ is the covariance matrix.

The value of $\mu_i$ and $\Sigma_i$ for a causal factor $Z_i$ can be computed using following formulas:

$$\mu_i = \sum_{j \in pa_i} \boldsymbol{f_W}_{i,j}(Y)\mu_j + \beta_{i,1}(Y), \tag{14}$$

$$\Sigma_{i,i} = \sum_{j \in pa_i} \boldsymbol{f_W}_{i,j}^2(Y)\Sigma_{j,j} + \beta_{i,2}(Y), \tag{15}$$

$$\Sigma_{i,j} = \sum_{k \in pa_i} \boldsymbol{f_W}_{k,j}(Y)\Sigma_{j,k}, for \ i \neq j, \tag{16}$$

where $pa_i$ represents the parents of the causal factor $Z_i$.

The associations among the causal factors can be represented by a fully connected DAG, which can be represented by an upper triangular matrix with zero-value diagonal elements. The weights of the associations among the causal factors can be generated through the learnable MLP $\boldsymbol{M}(Y)$. The Eq. 13 can be reformulated as

$$p_{\boldsymbol{f_W},\boldsymbol{\beta}}(\boldsymbol{Z}|Y) = p_{\boldsymbol{f_W},\boldsymbol{\beta}}(Z_1|Y) \prod_{i=2}^{m} p_{\boldsymbol{f_W},\boldsymbol{\beta}}(Z_i|\boldsymbol{Z}_{<i},Y) = \prod_{i=1}^{m} \mathcal{N}(\mu_{Z_i}, \sigma_{Z_i}^2), \tag{17}$$

where $\mu_{Z_i} = \sum_{j<i} W_{j,i}(Y)Z_j + \beta_{i,1}(Y), \sigma_{Z_i}^2 = \beta_{i,2}(Y)$, $m$ is the number of causal factors.

The corresponding inference model $q_\varphi(\boldsymbol{Z}|\boldsymbol{X}, Y)$ can be expressed as

$$q_\varphi(\boldsymbol{Z}|\boldsymbol{X},Y) = q_\varphi(Z_1|Y) \prod_{i=2}^{m} q_\varphi(Z_i|\boldsymbol{Z}_{<i},Y) = \prod_{i=1}^{m} \mathcal{N}(\mu'_{Z_i}, \sigma'^2_{Z_i}), \tag{18}$$

where $\mu'_{Z_i} = \sum_{j<i} W'_{j,i}(Y)Z_j + \beta'_{i,1}(Y), \sigma'^2_{Z_i} = \beta'_{i,2}(Y)$.

**Select examples via Co-Training manner** To prevent selection bias, we use two classification networks to select clean examples for each other. Correspondingly, we have two encoders, two instance decoders, two noisy label decoders, and two weight models.

First, two classification networks are used to output the clean labels:

$$Y_1 = \hat{q}_\psi^1(\boldsymbol{X}), Y_2 = \hat{q}_\psi^2(\boldsymbol{X}).$$

Then, we have two encoder networks to infer the causal factors based on the instances and clean labels:

$$\boldsymbol{Z_1} \sim q_\varphi^1(\boldsymbol{X}, Y_1), \boldsymbol{Z_2} \sim q_\varphi^2(\boldsymbol{X}, Y_2).$$

We also have four decoder networks to reconstruct instances and noisy labels based on the inferred causal factors:

$$\boldsymbol{X_1} = p_{\boldsymbol{f_X}}^1(\boldsymbol{Z_1}), \boldsymbol{X_2} = p_{\boldsymbol{f_X}}^2(\boldsymbol{Z_2}),$$
$$\tilde{Y}^1 = p_{\boldsymbol{f_{\tilde{Y}}}}^1(\boldsymbol{Z_1}), \tilde{Y}^2 = p_{\boldsymbol{f_{\tilde{Y}}}}^2(\boldsymbol{Z_2}).$$

The algorithm of the proposed method is shown in Alg. 1.

---

**Algorithm 1** GenP

---

**Input:** A noisy dataset $\tilde{\mathcal{D}}$, Total epoch $T_{max}$ .
1: $q_\psi^1(Y|\boldsymbol{X}), q_\psi^2(Y|\boldsymbol{X}) \leftarrow \text{WarmUP}(\tilde{\mathcal{D}})$;
2: **For** T $= 1, \ldots, T_{max}$:
3:    $\mathcal{S}_X, \mathcal{S}_U \leftarrow \text{Selection}(\tilde{\mathcal{D}}, q_\psi^1, q_\psi^2)$;
4:    $\mathcal{S}_X', \mathcal{S}_U' \leftarrow \text{MixUp}(\mathcal{S}_X, \mathcal{S}_U)$;
5:    **For** k$=1, 2$:
6:       Sample $(\boldsymbol{x}, \tilde{y}) \sim \tilde{\mathcal{D}}$;
7:       Sample $\hat{y} \sim q_\psi^k(\boldsymbol{x})$ via gumbel softmax;
8:       Sample $\boldsymbol{z} \sim q_\phi^k(\boldsymbol{x}, \hat{y})$;
9:       Feed $\hat{y}$ to the encoder $p_{\boldsymbol{f_W}, \boldsymbol{\beta}}^k$ to get the prior $p_{\boldsymbol{f_W}, \boldsymbol{\beta}}^k(\boldsymbol{Z}|Y = \hat{y})$;
10:      $\hat{x} \leftarrow p_{\boldsymbol{f_X}}^k(\boldsymbol{z})$;
11:      $\hat{\tilde{y}} \leftarrow p_{\boldsymbol{f_{\tilde{Y}}}}^k(\boldsymbol{z})$;
12:      Calculate the loss using Eq. 11 and update networks;
   **Output:** The classification networks $q_\psi^1(\cdot), q_\psi^2(\cdot)$.

---

**Optimization.** For experiments on Fashion-MNIST, CIFAR-10, CIFAR-100, and CIFAR-10N, we employed SGD with a momentum of 0.9 and a weight decay of 0.0005 to optimize the classification network. We used Adam with default parameters to optimize the encoder, weight model, instance decoder, and noisy label decoder. Our network was trained for 300 epochs with a batch size of 64. The initial learning rate for SGD was set at 0.02 and for Adam at 0.001. Both learning rates were reduced by a factor of 10 after 150 epochs. For experiments on Clothing1M, we employed SGD with a momentum of 0.9 and a weight decay of 0.001 to optimize the classification network. We used Adam with default parameters to optimize the encoder, weight model, instance decoder, and noisy label decoder. Our network was trained for 80 epochs with a batch size of 32. The initial learning rate for SGD was set at 0.002 and for Adam at 0.001. Both learning rates were reduced by a factor of 10 after 40 epochs.

## C   IDENTIFIABILITY ANALYSIS

The generative process can be defined as:

$$
\begin{aligned}
p_{\boldsymbol{f}}(\boldsymbol{X}, \tilde{Y}|\boldsymbol{Z}) &= p_{\boldsymbol{f_X}}(\boldsymbol{X}|\boldsymbol{M_X} \odot \boldsymbol{Z}) p_{\boldsymbol{f_{\tilde{Y}}}}(\tilde{Y}|\boldsymbol{M_{\tilde{Y}}} \odot \boldsymbol{Z}) \\
&= p_{\boldsymbol{\varepsilon_X}}(\boldsymbol{X} - \hat{p}_{\boldsymbol{f_X}}(\boldsymbol{Z})) p_{\boldsymbol{\varepsilon_{\tilde{Y}}}}(\tilde{Y} - \hat{p}_{\boldsymbol{f_{\tilde{Y}}}}(\boldsymbol{Z})),
\end{aligned}
\tag{19}
$$

which means that the value of $\boldsymbol{X}$ and $\tilde{Y}$ can be decomposed as $\boldsymbol{X} = \hat{p}_{\boldsymbol{f_X}}(\boldsymbol{Z}) + \boldsymbol{\varepsilon_X}, \tilde{Y} = \hat{p}_{\boldsymbol{f_{\tilde{Y}}}}(\boldsymbol{Z}) + \varepsilon_{\tilde{Y}}$, where $\boldsymbol{\varepsilon_X}$ and $\varepsilon_{\tilde{Y}}$ are independent noises variable with probability density function $p_{\boldsymbol{\varepsilon_X}}(\boldsymbol{\varepsilon_X})$ and $p_{\boldsymbol{\varepsilon_{\tilde{Y}}}}(\varepsilon_{\tilde{Y}})$.

Intuitively, the instances are generated by a subset of causal factors, and the noisy labels are generated by another subset of causal factors. When both the mixing functions $\hat{p}_{\boldsymbol{f_X}}(\cdot)$ and $\hat{p}_{\boldsymbol{f_{\tilde{Y}}}}(\cdot)$ are bijective, and the union of two subsets contains all causal factors, the function $\hat{p}_{\boldsymbol{f}}$ used to generate $\boldsymbol{X}$ and $\tilde{Y}$ is bijective.

The distribution of latent noise factors and causal factors can be reformulated to the exponential family, with the corresponding parameter $\boldsymbol{\eta_N}(Y)$ and $\boldsymbol{\eta_Z}(Y)$ respectively. Let $m$ denote the dimension of the sufficient statistics for the latent noise variables; $m$ also is the number of causal factors. Let $k$ denote the dimension of the sufficient statistics for the causal factors. Under the setting of this paper, $p_{\boldsymbol{f_W}, \boldsymbol{\beta}}(\boldsymbol{Z}|Y)$ is Gaussian, we have $k = m + (m(m+1))/2$. In our model, the number of causal factors is 4, thus $k = 14$. If we further assume $\beta_{i,2}(Y) = 1$, we have $k = (m(m+1))/2$. In our model, the number of causal factors is 4, thus $k = 10$ here. Then, we have the following theorem.

**Theorem 1.** *(Liu et al., 2022b) Suppose latent causal factors $\mathbf{Z}$ and the observed variables $Y, \tilde{Y}$ follow the generative model defined in Eq. 9 with parameters $(\boldsymbol{f}, \boldsymbol{f_W}, \boldsymbol{\beta})$. Assume the following holds:*

(i) *The set $\{\boldsymbol{x} \in \mathcal{X} | \varphi_{\boldsymbol{\varepsilon}}(\boldsymbol{x})\}$ has measure zero, where is the characteristic function of the density $p_{\boldsymbol{\varepsilon}}$.*

(ii) *The function $\hat{p}_{\boldsymbol{f}}$ is bijective.*

(iii) *There exist $2m + 1$ distinct points $\boldsymbol{y_{N0}}, \boldsymbol{y_{N1}}, \ldots, \boldsymbol{y_{N2m}}$, such that the matrix*
$$\boldsymbol{L_N} = (\boldsymbol{\eta_N}(Y = \boldsymbol{y_{N1}}) - \boldsymbol{\eta_N}(Y = \boldsymbol{y_{N0}}), \ldots, \boldsymbol{\eta_n}(Y = \boldsymbol{y_{N2m}}) - \boldsymbol{\eta_N}(Y = \boldsymbol{y_{N0}})) \tag{20}$$
*of size $2m \times 2m$ is invertible.*

(iv) *There exist $k + 1$ distinct points $\boldsymbol{y_{Z0}}, \boldsymbol{y_{Z1}}, \ldots, \boldsymbol{y_{Zk}}$, such that the matrix*
$$\boldsymbol{L_Z} = (\boldsymbol{\eta_Z}(Y = \boldsymbol{y_{Z1}}) - \boldsymbol{\eta_Z}(Y = \boldsymbol{y_{Z0}}), \ldots, \boldsymbol{\eta_Z}(Y = \boldsymbol{y_{Zk}}) - \boldsymbol{\eta_z}(Y = \boldsymbol{y_{Z0}})) \tag{21}$$
*of size $k \times k$ is invertible.*

(v) *The function class of $f_W(\cdot)_{i,j}$ can be expressed by a Taylor series: for each $f_W(\cdot)_{i,j}$, $f_W(\mathbf{0})_{i,j} = 0$.*

*then the true latent causal variables $\mathbf{Z}$ are related to the estimated latent causal variables $\hat{\mathbf{Z}}$ by the following relationship: $\mathbf{Z} = \boldsymbol{P}\hat{\mathbf{Z}} + \boldsymbol{c}$, where $\boldsymbol{P}$ denotes the permutation matrix with scaling, $\boldsymbol{c}$ denotes a constant vector.*

The theorem guarantees that the causal factors can be identifiable up to permutation and scaling under some mild assumptions. The Assumption (iii) requires $2m + 1$ distinct classes, and the Assumption (iv) requires $k + 1$ distinct classes. But it does not mean that we must have $2m + 1 + k + 1$ distinct classes. The $k + 1$ distinct classes that satisfy the Assumption (iv) could contain the $2m + 1$ distinct classes that satisfy the Assumption (iii). Ideally, we may only need $k + 1$ distinct classes to achieve identifiability. The supervision information in our work is from the labels in confident examples. Thus, the confident examples have to contain the data points to satisfy these conditions, *i.e.*, . When the number of causal factors is 4, it needs 15 distinct classes, *i.e.*, the class number is at least 15. If we further assume $\beta_{i,2}(Y) = 1$, it only needs 11 distinct classes.

In summary, when the number of distinct classes is larger than the sufficient statistics of the latent variable $\mathbf{Z}$, and the confident examples at least contain an example of each class. Then the variable $\mathbf{Z}$ could be identified.

## D    DIFFERENCES FROM PREVIOUS WORK

Previous research on learning with noisy labels has incorporated generative models (Yao et al., 2021; Garg et al., 2023). However, our approach distinguishes itself from these earlier methods.

Previous work helps the learning of classifiers by exploiting the information contained in the distribution of instances. To be specific, when the latent clean label $Y$ is a cause of the instance $\mathbf{X}$, the distribution of instance $p(\mathbf{X})$ will generally contain some information about the distribution of clean class posterior $p(Y|\mathbf{X})$. To exploit the information contained in the instances, CausalNL and InstanceGM use the generative model to model the generative relationship between the latent variable $\mathbf{Z}$ and the observed instance $\mathbf{X}$. However, they do not model the causal structure among the latent variable $\mathbf{Z}$. Therefore, their methods only "partially" model the noisy data generative process. These methods do not analyze the identifiability of the generative process.

Our aim is to enable deep neural networks to capture the connections of noise transitions among different instances automatically. Since once the connections are captured, the noise transition learned in some examples can be generalized to other examples. To achieve this, we need to model the joint distribution of all variables $p(\mathbf{X}, \tilde{Y}, Y, \mathbf{Z}, \mathbf{N})$. The noise transition can be obtained through

$$p(\tilde{Y}|\mathbf{X}, Y) = \frac{\int_{\mathbf{Z},\mathbf{N}} p(\mathbf{X}, \tilde{Y}, Y, \mathbf{Z}, \mathbf{N}) d\mathbf{Z} d\mathbf{N}}{\int_{\tilde{Y},\mathbf{Z},\mathbf{N}} p(\mathbf{X}, \tilde{Y}, Y, \mathbf{Z}, \mathbf{N}) d\tilde{Y} d\mathbf{Z} d\mathbf{N}}.$$

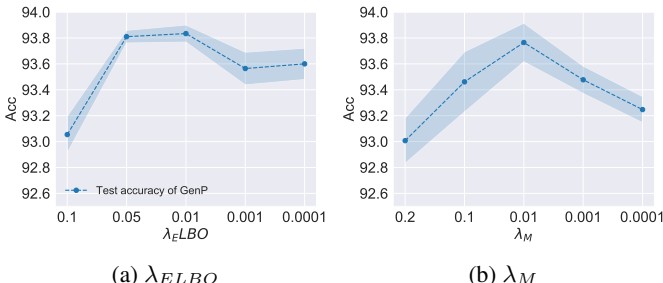

(a) $\lambda_{ELBO}$            (b) $\lambda_M$

Figure 4: Illustration of the test accuracy on CIFAR-10N with noise type "worst". The error bar for standard deviation has been shaded.

Table 6: The comparison of noise transition estimation error on CIFAR-10.

| | CIFAR-10 | | | | |
|---|---|---|---|---|---|
| | IDN-10% | IDN-20% | IDN-30% | IDN-40% | IDN-50% |
| BLTM | $0.275 \pm 0.022$ | $0.339 \pm 0.020$ | $0.467 \pm 0.008$ | $0.670 \pm 0.106$ | $0.816 \pm 0.056$ |
| MEIDTM | $0.240 \pm 0.002$ | $0.374 \pm 0.001$ | $0.563 \pm 0.001$ | $0.775 \pm 0.002$ | $0.985 \pm 0.001$ |
| VolMinNet | $0.477 \pm 0.005$ | $0.590 \pm 0.000$ | $0.680 \pm 0.000$ | $0.720 \pm 0.000$ | $0.783 \pm 0.005$ |
| GenP | $\mathbf{0.224 \pm 0.005}$ | $\mathbf{0.302 \pm 0.016}$ | $\mathbf{0.374 \pm 0.008}$ | $\mathbf{0.434 \pm 0.021}$ | $\mathbf{0.548 \pm 0.041}$ |

To obtain this joint distribution, we propose a principled way to model the whole data generative process. Specifically, we not only model the generative process from the latent variable $\mathbf{Z}$ to the instance $\mathbf{X}$ but also model and learn causal structure among different latent variables $\{Z_1, Z_2, \ldots, Z_m\}$ from observed noisy data.

## E    HYPER-PARAMETER SENSITIVITY

We analyze the sensitivity regarding hyperparameters ($\lambda_{ELBO}$ and $\lambda_M$) of the model on the CIFAR-10N dataset, and the noise type is "worst". The experiment results are shown in Fig. 4. The hyperparameters $\lambda_{ELBO}$ and $\lambda_M$ are set as 0.01.

## F    NOISE TRANSITION ESTIMATION ERROR

We conduct experiments on the dataset CIFAR-10 to calculate the noise transition estimation error. The labels are corrupted manually using the instance-dependent noisy label generation method proposed in Xia et al. (2020). The noise rates are from 0.1 to 0.5. The baselines used to compare the noise transition estimation error are BLTM (Yang et al., 2022), MEIDTM (Cheng et al., 2022a) and VolMinNet (Li et al., 2021). The experiment results are shown in Tab. 6. The experiment results indicate that the proposed method outperforms these baselines in estimating noise transition.

## G    COMPARISON OF THE NOISE TRANSITION

We use t-SNE visualization to compare the noise transition inferred by our method with those derived from the MEIDTM (Cheng et al., 2022a). We also select 30 pairs of data points with the same predicted clean labels. The dataset is CIFAR-10 with instance-dependent label noise, and the noise rate is 50%. The experiment results are shown in Fig. 5. These data points are the same in two figures. We can see that the distance between the same pair is different in the two images. For example, the pairs with number 21 are close to each other in the first figure but are further apart in the second figure. This can verify that the similarity inferred by our method is different from the instance-dependent transition matrix-based method MEIDTM.

Table 7: Means and standard deviations (percentage) of classification accuracy on CIFAR-10N.

| | Worst | Aggregate | Random 1 | Random 2 | Random 3 |
|---|---|---|---|---|---|
| | CIFAR-10N | | | | |
| GenP (alternative) | $93.72 \pm 0.07$ | $95.21 \pm 0.19$ | $95.14 \pm 0.12$ | $95.28 \pm 0.09$ | $94.99 \pm 0.15$ |
| GenP (end-to-end) | $\mathbf{93.87 \pm 0.13}$ | $\mathbf{95.39 \pm 0.18}$ | $\mathbf{95.38 \pm 0.13}$ | $\mathbf{95.30 \pm 0.12}$ | $\mathbf{95.26 \pm 0.13}$ |

Table 8: Means and standard deviations (percentage) of classification accuracy on CIFAR-10N.

| | Worst | Aggregate | Random 1 | Random 2 | Random 3 |
|---|---|---|---|---|---|
| | CIFAR-10N | | | | |
| CE on the selected dataset | $92.13 \pm 0.05$ | $91.64 \pm 0.03$ | $90.94 \pm 0.07$ | $90.06 \pm 0.04$ | $89.10 \pm 0.25$ |
| GenP | $\mathbf{93.87 \pm 0.13}$ | $\mathbf{95.39 \pm 0.18}$ | $\mathbf{95.38 \pm 0.13}$ | $\mathbf{95.30 \pm 0.12}$ | $\mathbf{95.26 \pm 0.13}$ |

## H  ABLATION STUDY

**Comparison between end-to-end learning and alternative learning approaches**  We conduct the experiments of the alternative learning approach, which optimizes $\mathcal{L}_{semi}$ and $-\lambda_{ELBO}ELBO + \lambda_M(\|\boldsymbol{M_X}\|_1 + \|\boldsymbol{M_{\tilde{Y}}}\|_1)$ alternatively. The dataset is CIFAR-10N. The experiment results are shown in Tab. 7. Empirically, the performance of the end-to-end learning approach is better than the alternative learning approach.

**Ablation study based on the number of latent variables**  We conduct the ablation study based on the number of latent variables on the CIFAR-10N dataset, and the noise type is "worst". The experiment results are shown in Fig. 6. In our experiment, the number of latent variables is set as 4.

## I  THE DISTRIBUTION OF SELECTED EXAMPLES

We manually select all the examples with correct labels on the noisy dataset. These examples can be viewed as a perfectly selected dataset.

To explore the difference between the distribution of selected examples and the distribution of examples in the clean domain, we first use t-SNE to visualize the selected examples. The selected examples are plotted in red, and other examples in the dataset are plotted in gray. The dataset is CIFAR-10 with instance-dependent label noise, and the noise rate is 0.5. The visualized result is shown in Fig. 7. As shown in the Fig. 7, the distribution of the selected dataset is different from the whole dataset.

We then train a classifier on the selected examples with standard cross-entropy loss. The noise rates are 0.1, 0.2, 0.3, 0.4, and 0.5. The experiment results are shown in Tab. 8. The experiment results show that the performance of the method using standard cross entropy loss on the selected examples is lower than our method, even if the classifier is trained on a perfect elected dataset.

## J  RECONSTRUCTED IMAGE VISUALIZATION

We visualize the reconstructed images from our model trained on the FashionMNIST dataset with a noise rate of 0.5. Due to time constraints, the training epoch is 30, whereas the training epoch is 300 in other experiments. The visualization results are shown in Fig. 8. The images are arranged in a comparative format: the first and third columns display the original images, while the second and fourth columns show the corresponding reconstructed images by the proposed model, alternating in this pattern throughout the display. The experiment results demonstrate that the model can successfully reconstruct images.

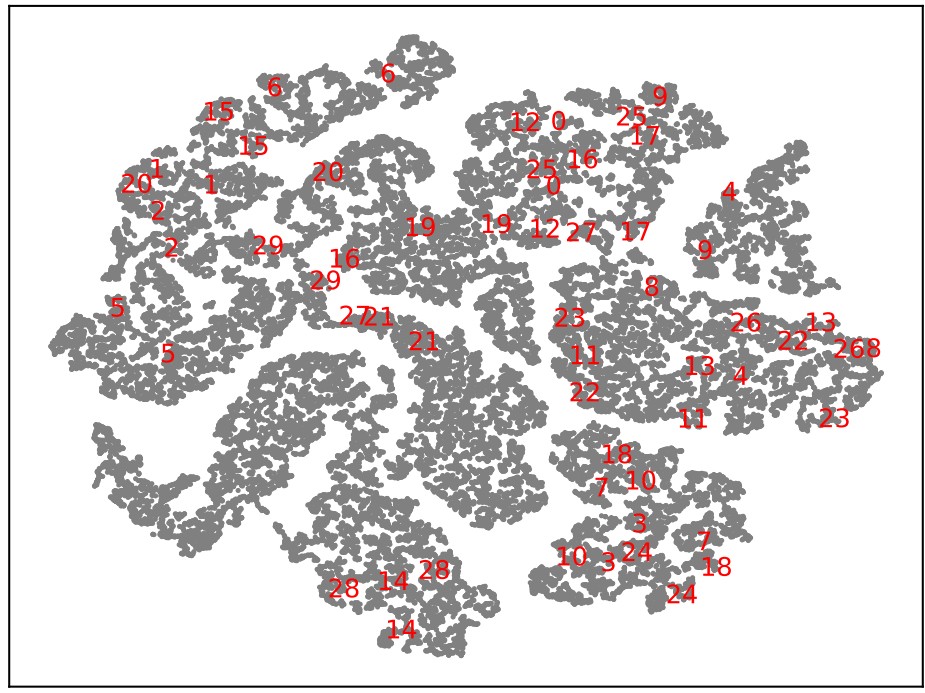

(a) GenP

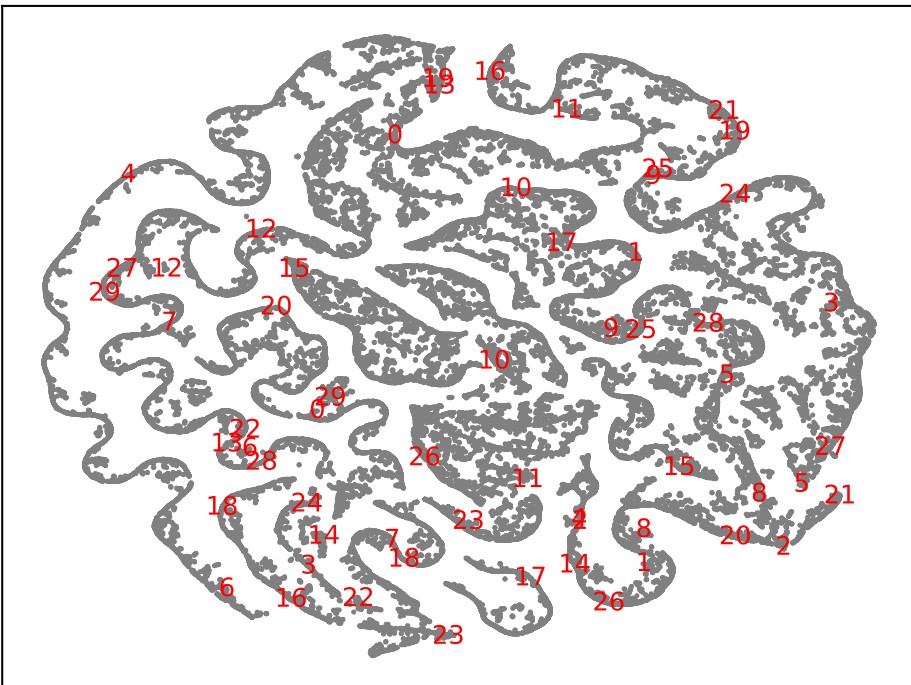

(b) MEIDTM

Figure 5: The t-SNE visualization of the similarity of the learned noise transition. The pairs of data points with the same predicted clean label are marked with the same number. The distances between two data points represent the difference between the two noise transitions of these data points. The distance between the pair with the same number is different in the two images.

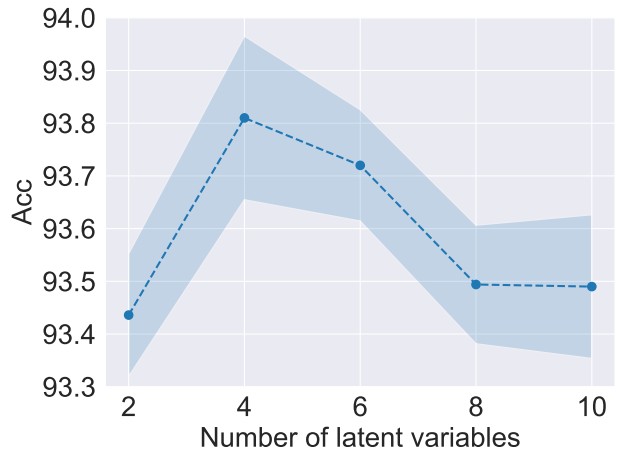

Figure 6: The ablation study based on the number of latent variables.

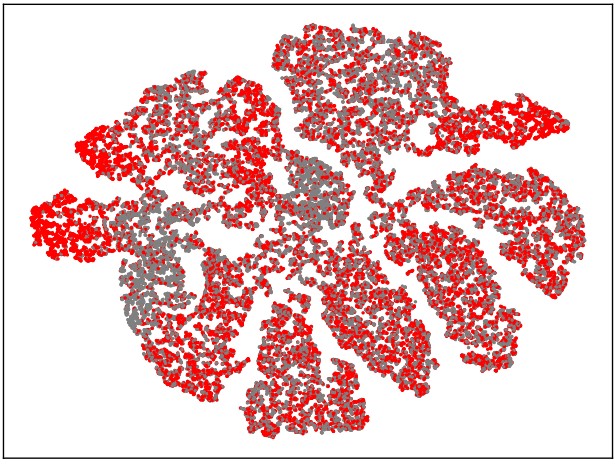

Figure 7: The t-SNE visualization of the selected examples.

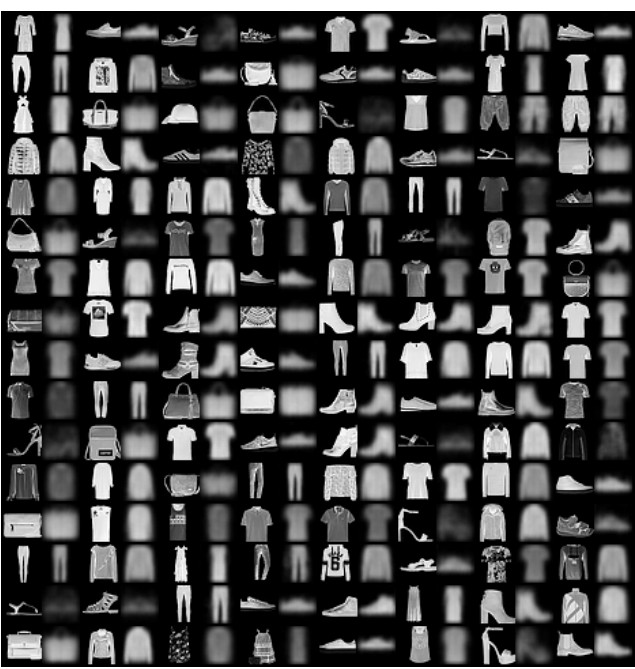

Figure 8: The visualization of the reconstructed images.

