# OpenReview forum: "Learning the Latent Noisy Data Generative Process for Label-Noise Learning"
_ICLR.cc/2024/Conference — Submitted to ICLR 2024_

### Official Review · Reviewer_XDR1 · 2023-10-15

**Soundness:** 3 good
**Presentation:** 4 excellent
**Contribution:** 1 poor
**Rating:** 3
**Confidence:** 5

**Summary:**

Although inferring noise transition is crucial, its inference is limited from assumptions of the relations between instances and its respective noise transitions. This relation should be trained as latent (by learning noisy data generative process). This paper learns this relation by introducing graph structure. Empirically, it shows good performance.

**Strengths:**

- The motivation which underlines the problems of the previous studies, the assumption of the similarity of noise transitions, is adequate.
- The structure is flexible.

**Weaknesses:**

- To the best of my knowledge, BLTM (one of the baselines in the experiment part), which trains a instance-dependent transition matrix network using the distilled dataset, has no assumption for the similarity between noise transition for different instances. I think this study contradicts to author's proposal, which says, there are assumptions of similarity in the previous researches.
- Since Y is latent, there is no way to prove that Y is true (so that the authors relied on distillation). Therefore, it means the authors actually makes an another assumption saying that the distilled dataset is clean.
- Since it should generate X (usually an image), time complexity and memory will be large. Why should we generate an image to classify? Also want to see the exact time and memory comparison over several baselines.
- Lack of any analysis. In the experiment part, nothing but the test accuracy are reported. Please show analytical empirical results for whether the training are processed as authors proposed. For example, at least whether the error of the inferred transition is small or not should be expressed.

**Questions:**

- Can authors show (at least, the empirical) evidences of the proposal saying that "noise transitions can be inferred of a special group of instances"?
- Why should we use graph structure? Can't this structure be changed to other structure?
- How can we be sure that the process the authors proposed improve the model performance?
- Why using the estimated clean labels from distillation is not enough? As the authors said, distillation methods are well studied. By pointing out that the model performance of the method the authors proposed should depend on the quality of the estimated clean labels, the distilled labels should be clean enough. If the estimated labels are clean enough, why should we do more process fundamentally?

---

> ### Author Response · Authors · 2023-11-18
> **Response to Reviewer XDR1**
>
> We sincerely thank you for your time and effort in reviewing our manuscript and providing valuable comments. Please find the response to your comments below.
>
> **Q1: BLTM has no assumption for the similarity between noise transition for different instances, thus the study on BLTM contradicts the author's proposal.**
>
> **A1**: BLTM can learn the noise transition of some examples using a neural network. The authors of BLTM hope the network can generate the noise transition for unseen instances. However, the author does not discuss how to generalize the learned noise transition to other examples, and they do not provide a theoretical guarantee that the noise transition can be generalized. Some hidden assumptions must exist to build the connection. Otherwise, the noise transition can not be generalized to other examples. Our claim focuses on the methods that have theoretical guarantees on the generalization of noise transitions, e.g., the noise transition is class-dependent, and the noise transition for the instances in the same manifold is the same. To avoid confusion, we have modified the paper.
>
> We are the first to explore when the learned noise transition in some instances can be generalized to other instances in learning with noisy labels. Our aim is to enable deep neural networks to capture the connections of noise transitions among different instances automatically. Since once the connections are captured, the noise transition learned in some examples can be generalized to other examples. To achieve this, we need to model the joint distribution of all variables $p(\mathbf{X},\tilde{Y},Y,\mathbf{Z},\mathbf{N})$. The noise transition can be obtained through
> $$
> p(\tilde{Y}|\mathbf{X},Y)=\frac{\int_{\mathbf{Z},\mathbf{N}}p(\mathbf{X},\tilde{Y},Y,\mathbf{Z},\mathbf{N})d\mathbf{Z}d\mathbf{N}}{\int_{\tilde{Y},\mathbf{Z},\mathbf{N}}p(\mathbf{X},\tilde{Y},Y,\mathbf{Z},\mathbf{N})d\tilde{Y}d\mathbf{Z}d\mathbf{N}}.
> $$
> To obtain this joint distribution, we propose a principled way to model the whole data generative process. Specifically, we not only model the generative process from the latent variable $\mathbf{Z}$ to the instance $\mathbf{X}$ but also model and learn causal structure among different latent variables $\{Z_1,Z_2,\dots,Z_m\}$ from observed noisy data.
>
> To achieve this, the generation of the variable $\mathbf{N}$ is modeled by a Gaussian distribution. The generation of the latent variable $\mathbf{Z}$ is modeled by a Structure Causal Model (SCM): $Z_i:=\mathbf{W}^T_i \mathbf{Z}+N_i$, where $\mathbf{W}$ is a matrix used to represent the association among the latent variables. Two generators are used to model the generation of the instance $\mathbf{X}$ and the noisy label $\tilde{Y}$, respectively.
>
> Moreover, our paper has analyzed the identifiability of the proposed method theoretically in Appendix C. We discuss the conditions required for identifiability. Specifically, when the number of distinct classes is large (more than the sufficient statistics of the latent variable $\mathbf{Z}$) and we have a confident example in each class, the latent variable $\mathbf{Z}$ can be identified up to permutation.
>
> **Q2: The authors actually make another assumption, saying that the selected dataset is clean.**
>
> **A2**: We assume that the selected dataset is likely clean. This assumption is reasonable because the accuracy of the selected dataset is high empirically. For example, when the dataset is CIFAR-10, and the noise type is "worst", the accuracy of the selected dataset is 0.9568.
>
> It is also worth mentioning that without clean labels, recovering the causal factor $\mathbf{Z}$ is an ambitious goal and generally impossible. This is because the nonlinear mixture function $f(\cdot)$ used to generate observed variable $\mathbf{X}$ is unknown. A variable $\mathbf{Z}'$ can be constructed via a nonlinear function $\mathbf{Z}'=g(\mathbf{Z})$. Another nonlinear mixture function $f'(\cdot)$ can generate the same observation, i.e., $\mathbf{X}=f'(\mathbf{Z}')$. Previous studies also prove that it is ill-posed to recover the latent causal factors with only the observed variables [1,2]. Existing methods in causal representation learning also have to use the auxiliary variables to recover the causal factors [3,4]. In the context of learning with noisy labels, auxiliary variables necessitate clean labels.
>
> **Q3: Time complexity and memory will be large.**
>
> **A3**: The proposed method does not increase the time complexity and memory because the encoder and the decoder introduced by our method are lightweight. The training time of the proposed method on CIFAR-10 is 13.68 hours, and the training time of DivideMix is 9.08 hours. The memory used for the proposed method is 7.06 GB, and the memory used for DivideMix is 6.99 GB.
>
> Moreover, the encoder and decoder can be discarded during inference. The inference time is the same as the time of baselines.

---

> ### Author Response · Authors · 2023-11-18
> **Response to Reviewer XDR1 (Continue)**
>
> **Q4: Whether the error of the inferred transition is small or not should be expressed.**
>
> **A4**: We have conducted experiments on the dataset CIFAR-10 to calculate the noise transition estimation error. The labels are corrupted manually using the instance-dependent noisy label generation method proposed in [5]. The baselines used to compare the noise transition estimation error are BLTM, MEIDTM and VolMinNet. The experiment results are shown as follows.
>
> |           | IDN-10%               | IDN-20%               | IDN-30%               | IDN-40%               | IDN-50%               |
> | --------- | --------------------- | --------------------- | --------------------- | --------------------- | --------------------- |
> | BLTM      | 0.275 $\pm$ 0.022     | 0.339 $\pm$ 0.020     | 0.467 $\pm$ 0.008     | 0.670 $\pm$ 0.106     | 0.816 $\pm$ 0.056     |
> | MEIDTM    | 0.240 $\pm$ 0.002     | 0.374 $\pm$ 0.001     | 0.563 $\pm$ 0.001     | 0.775 $\pm$ 0.002     | 0.985 $\pm$ 0.001     |
> | VolMinNet | 0.477 $\pm$ 0.005     | 0.590 $\pm$ 0.000     | 0.680 $\pm$ 0.000     | 0.720 $\pm$ 0.000     | 0.783 $\pm$ 0.005     |
> | GenP      | **0.224 $\pm$ 0.005** | **0.302 $\pm$ 0.016** | **0.374 $\pm$ 0.008** | **0.434 $\pm$ 0.021** | **0.548 $\pm$ 0.041** |
>
> Empirical results show that our method can reduce the error of inferred noise transition.
>
> **Q5: Why should we use graph structure?**
>
> **A5**: To model the causal structure among latent variable $\mathbf{Z}$, the graph structure has to be used. This is because the causal structure among $\mathbf{Z}$ is a part of the noisy data generative process. As mentioned in A1, modeling the noisy data generative process can obtain the joint distribution $p(\mathbf{X},\tilde{Y},Y,\mathbf{Z},\mathbf{N})$, then the connections of noise transitions among different instances can be captured automatically. The more accurately the generation process is modeled, the more accurate the joint distribution obtained will be.
>
> **Q6: Evidence of the proposal saying that "noise transitions can be inferred of a special group of instances"?**
>
> **A6**: The evidence is the noise transition estimation methods in previous work. For example, the special group of instances can be the anchor points in the dataset, which is stated in Section 2 in [6] and Theorem 3 in [7].
>
> **Q7: If the estimated labels are clean enough, why should we fundamentally do more processes?**
>
> **A7:** The size of selected examples is much smaller than the size of the whole dataset, and the distribution of selected examples is not the same as the distribution of examples in the clean domain. As a result, training classifiers on selected examples with different distributions will lead to the performance degradation of the classifiers because the parameters of the learned classifier are not statistically consistent with the parameters of the optimal classifier defined on the clean domain.
>
> To guarantee the statistical consistency of classifiers, the information contained in the remaining examples has to be used. Modeling the noise transition for the whole dataset is a way to employ the information contained in the training examples. The reason is that the clean class posterior $p(\tilde{Y}|\mathbf{X}=\mathbf{x})$ for an instance $\mathbf{x}$ can be inferred through the noise transition $p(\tilde{Y}|Y,\mathbf{X}=\mathbf{x})$ and the noisy class posterior $p(\tilde{Y}|\mathbf{X}=\mathbf{x})$, i.e.,  $p(\tilde{Y}|\mathbf{X}=\mathbf{x})=p(\tilde{Y}|Y,\mathbf{X}=\mathbf{x})p(Y|\mathbf{X}=\mathbf{x})$.

---

> ### Author Response · Authors · 2023-11-18
> **Response to Reviewer XDR1 (Continue)**
>
> **Q8: How can we be sure that the process the authors proposed improves the model performance?**
>
> **A8**: Our method is conceptually better than other methods. As discussed in the Introduction, to generalize the noise transition learned in some instances to other instances, existing work without theoretical guarantee manually defines similarity for noise transition across various instances. However, the human-defined similarity is hard to verify in the real world. If the similarity is not truthful, the estimation error of the noise transition could be large. Thus, the performance of the classifier would decrease.
>
> To avoid human-defined similarity, we propose a principled way to capture the connections of noise transitions among different instances automatically. Specifically, our method models the data generative process, and then the joint distribution $p(\mathbf{X},\tilde{Y},Y,\mathbf{Z},\mathbf{N})$ can be obtained. Thus, the noise transition can be captured  (please refer to the answer in A1). Once the model can capture the connections, the noise transition estimated in some examples can be generalized to other examples. The information contained in other examples can be inferred and learned by the model. Thus, the performance of the model will be improved. The experiment results also have verified the effectiveness of our method.
>
> ### References
>
> [1] Hyvärinen, Aapo, and Petteri Pajunen. "Nonlinear independent component analysis: Existence and uniqueness results." *Neural networks* 12.3 (1999): 429-439.
>
> [2] Locatello, Francesco, et al. "Challenging common assumptions in the unsupervised learning of disentangled representations." *international conference on machine learning*. PMLR, 2019.
>
> [3] Yang, Mengyue, et al. "Causalvae: Disentangled representation learning via neural structural causal models." *Proceedings of the IEEE/CVF conference on computer vision and pattern recognition*. 2021.
>
> [4] Liu, Yuhang, et al. "Identifying weight-variant latent causal models." *arXiv preprint arXiv:2208.14153* (2022).
>
> [5] Xia, Xiaobo, et al. "Part-dependent label noise: Towards instance-dependent label noise." *Advances in Neural Information Processing Systems* 33 (2020): 7597-7610.
>
> [6] Xia, Xiaobo, et al. "Are anchor points really indispensable in label-noise learning?." *Advances in neural information processing systems* 32 (2019).
>
> [7] Patrini, Giorgio, et al. "Making deep neural networks robust to label noise: A loss correction approach." *Proceedings of the IEEE conference on computer vision and pattern recognition*. 2017.

---

> ### Comment · Reviewer_XDR1 · 2023-11-20
> **Response to the authors**
>
> Thank authors for their sincere efforts to respond to my concerns. Still, my concerns below are not solved yet, and I require the authors to respond to those.
>
> 1. I want to know the number of samples and accuracies with regard to the distilled dataset for several noisy dataset settings to support the proposal of authors, saying "assume the selected dataset is likely clean is reasonable" in A2 and "the distribution of selected examples is not the same as the distribution of examples in the clean domain" in A7. Also want to know the performance comparison when trained only with distilled dataset, since if the peformance only with the distilled dataset is similar to the performance the authors proposed in the paper, I think it will take far less time and memory complexity.
>
> 2. If the distribution of selected examples is not the same as the distribution of examples in the clean domain, how can the author can assume the learned causal factor with distilled samples is trustworthy?
>
> 3. Time and memory comparison is not enough. Dividemix uses two networks, so it is so certain that this method would spend computation than other baselines. The time and memory comparison with regard to all other baselines, or at least with regard to other transition matrix based methods can tell how much time and memory is used usually and enable the evaluation of the method.
>
> Also, I think the time and memory gap could be larger when this method is applied to more complicated dataset, e.g. Clothing1M, than CIFAR10.
>
> 4. Can the authors specify how they calculated the transition matrix estimation error for the instance dependent transition matrix? e.g. how to measure the difference between the ground truth and the estimated transtion matrix, since the transition matrix should be different for every instance? / What is the meaning of estimation error of other dimensions when a true label of a sample is only one? e.g. the transition matrix error of label dimension 0,1, and 3~9 for 10-class classification when a true label of a sample is 2?
>
> 5. I found out the authors did not compare theirs with class-dependent transition matrix estimation based methods. However, I think it should be compared as baselines.
>
> 6. Additionally, I doubt the authors lack baselines. e.g. in [1] , which utilizes class dependent transition matrix, they achieved 75.12% for Clothing1M. However, this paper shows 74.81%, even with instance dependent transition.
>
> 7. I still cannot trust that the suggested method can significantly increae the resulting classifier's performance. I think this concern can be solved if the authors analyze what the graph structure tells about the relations between samples empirically, and how much different is the similarity inferred from the graph structure and human arbitrarily defined similarity (or currently modeled several instance-dependent transition matrix based methods).
>
> [1] Cheng, D., Ning, Y., Wang, N., Gao, X., Yang, H., Du, Y., ... & Liu, T. (2022). Class-Dependent Label-Noise Learning with Cycle-Consistency Regularization. Advances in Neural Information Processing Systems, 35, 11104-11116.

---

> > ### Author Response · Authors · 2023-11-22
> > **Response to Reviewer XDR1**
> >
> > Thank you for your valuable comments.
> >
> > **Q1: The empirical support of the selected dataset is likely clean.**
> >
> > **A1**:  We list the accuracy of the selected dataset here. The dataset is CIFAR-10. We can see that the accuracy is high. Thus, the selected dataset is likely clean.
> >
> > | IDN-10% | IDN-20% | IDN-30% | IDN-40% | IDN-50% |
> > | ------- | ------- | ------- | ------- | ------- |
> > | 99.54%  | 99.22%  | 99.12%  | 99.01%  | 98.46%  |
> >
> > **Q2: The performance comparison when trained only with a distilled dataset.**
> >
> > **A2**: We manually select all the examples with correct labels on the noisy dataset. These examples can be viewed as a perfectly distilled dataset. Then, we train a classifier on these examples using standard cross-entropy loss. The dataset used in our experiments is CIFAR-10.
> >
> > The performance of the classifier is shown as follows.
> >
> > |                            | IDN-10%              | IDN-20%              | IDN-30%              | IDN-40%              | IDN-50%              |
> > | -------------------------- | -------------------- | -------------------- | -------------------- | -------------------- | -------------------- |
> > | CE on the selected dataset | 92.13 $\pm$ 0.05     | 91.64 $\pm$ 0.03     | 90.94 $\pm$ 0.07     | 90.06 $\pm$ 0.04     | 89.10 $\pm$ 0.25     |
> > | GenP                       | **96.12 $\pm$ 0.12** | **96.05 $\pm$ 0.12** | **95.74 $\pm$ 0.13** | **95.44 $\pm$ 0.12** | **89.39 $\pm$ 0.45** |
> >
> > The results show that even if the classifier is trained on a perfectly distilled dataset, the performance of the method with standard cross-entropy loss is still lower than our method.
> >
> > **Q3: The empirical support of "the distribution of selected examples is not the same as the distribution of examples in the clean domain".**
> >
> > **A3**: To explore the difference between the distribution of selected examples and the distribution of examples in the clean domain, we use t-SNE to visualize the selected examples. The selected examples are plotted in red, and other examples in the dataset are plotted in gray. The dataset is CIFAR-10 with instance-dependent label noise, and the noise rate is 0.5. The visualized result is shown in Appendix I. As shown in the picture, the distribution of the selected dataset is different from the whole dataset.
> >
> > **Q4: How can the author assume the learned causal factor with distilled samples is trustworthy when the distribution of selected examples is not the same as the distribution of examples in the clean domain?**
> >
> > **A4**: Thanks for pointing out this insightful problem.
> >
> > We have explored the sufficient condition to identify causal factors in Appendix C. Specifically, when the number of distinct classes is large (more than the sufficient statistics of the latent variable $\mathbf{Z}$), and we have a confident example in each class, the latent variable $\mathbf{Z}$ can be identified up to permutation.
> >
> > In our experiment settings, we assume the number of latent variables is 4, and it needs 15 distinct classes to identify the causal factors. To identify these latent variables, the confident examples must contain the examples from these distinct classes, i.e., the distribution of confident examples must cover the distribution of the examples on these distinct classes. In the real world, the number of latent variables is much larger (There may be thousands of latent variables). To identify the latent variables, more fine-grained distinct classes have to be included. If the confident examples do not contain the needed distinct classes, the learned causal factor is not trustworthy. However, our method can still encourage the model to identify latent causal factors. In the experiments, the empirical results have verified the effectiveness of our method.

---

> > > ### Author Response · Authors · 2023-11-22
> > > **Response to Reviewer XDR1 (Continue)**
> > >
> > > **Q5: The time and memory comparison with regard to all other baselines.**
> > >
> > > **A5**: We show the time and memory of other baselines. We conduct the experiments on the CIFAR-10N dataset with noise type "worst". The overall training time (hours) of each algorithm is shown as follows.
> > >
> > > | CE      | MentorNet    | CoTeaching | Reweight      | Forward  |
> > > | ------- | ------------ | ---------- | ------------- | -------- |
> > > | 0.45    | 0.46         | 0.81       | 0.56          | 0.55     |
> > > | **PTD** | **CausalNL** | **MEIDTM** | **DivideMix** | **GenP** |
> > > | 1.15    | 2.71         | 0.36       | 9.08          | 13.68    |
> > >
> > > We also show the memory (GB) used when training the algorithms.
> > >
> > > | CE      | MentorNet    | CoTeaching | Reweight      | Forward  |
> > > | ------- | ------------ | ---------- | ------------- | -------- |
> > > | 3.07    | 3.07         | 4.95       | 3.83          | 4.72     |
> > > | **PTD** | **CausalNL** | **MEIDTM** | **DivideMix** | **GenP** |
> > > | 5.78    | 5.62         | 5.35       | 6.99          | 7.06     |
> > >
> > > Though the training time and memory of GenP are larger than those of other methods, modern computers can compute our algorithm easily. Meanwhile, the performance of our algorithm is better than other methods.
> > >
> > > **Q6: How they calculate the transition matrix estimation error for the instance-dependent transition matrix.**
> > >
> > > **A6**: We follow the previous work [6] to use the relative error to the transition matrix estimation error for each instance. Let $\hat{T}_i$ denote the estimated noise transition for the $i$-th clean label, and $T_i$ denote the ground truth. The transition error can be calculated through $|T_i-\hat{T}_i|/|T_i|$. We use the instance-dependent label noise generation method in [5]. The estimation error for the other label $j$, where $j \neq i$ is ignored because we do not have the ground truth $j$ to calculate the error.
> > >
> > > **Q7: I doubt the authors lack a baseline that utilizes the class class-dependent transition matrix.**
> > >
> > > **A7**: Thanks for your advice, we have added this baseline [7] in our paper. In our paper, we refer to this method as CCR. Here, we list the comparison results on CIFAR-10N.
> > >
> > > |      | Worst                | Aggregate            | Random 1             | Random 2             | Random 3             |
> > > | ---- | -------------------- | -------------------- | -------------------- | -------------------- | -------------------- |
> > > | CCR  | 80.43 $\pm$ 0.24     | 90.10 $\pm$ 0.09     | 88.53 $\pm$ 0.08     | 88.21 $\pm$ 0.11     | 88.46 $\pm$ 0.08     |
> > > | GenP | **93.87 $\pm$ 0.13** | **95.39 $\pm$ 0.18** | **95.38 $\pm$ 0.13** | **95.30 $\pm$ 0.12** | **95.26 $\pm$ 0.13** |
> > >
> > > **Q8: Compare with class-dependent transition matrix-based methods.**
> > >
> > > **A8**: We actually have compared with class-dependent transition matrix-based methods in our paper: Reweight and Forward. Now, we add a new class-dependent transition matrix-based method proposed in [7]. Here, we list the comparison results of Reweight, Forward, CCR and GenP on CIFAR-10N.
> > >
> > > |          | Worst                | Aggregate            | Random 1             | Random 2             | Random 3             |
> > > | -------- | -------------------- | -------------------- | -------------------- | -------------------- | -------------------- |
> > > | Reweight | 77.68 $\pm$ 2.46     | 89.34 $\pm$ 0.09     | 88.44 $\pm$ 0.10     | 88.16 $\pm$ 0.10     | 88.03 $\pm$ 0.10     |
> > > | Forward  | 79.27 $\pm$ 1.18     | 89.22 $\pm$ 0.21     | 86.84 $\pm$ 0.97     | 86.99 $\pm$ 0.10     | 87.53 $\pm$ 0.34     |
> > > | CCR      | 80.43 $\pm$ 0.24     | 90.10 $\pm$ 0.09     | 88.53 $\pm$ 0.08     | 88.21 $\pm$ 0.11     | 88.46 $\pm$ 0.08     |
> > > | GenP     | **93.87 $\pm$ 0.13** | **95.39 $\pm$ 0.18** | **95.38 $\pm$ 0.13** | **95.30 $\pm$ 0.12** | **95.26 $\pm$ 0.13** |
> > >
> > > **Q9: How much different is the similarity inferred from the graph structure and human arbitrarily defined similarity (or currently instance-dependent transition matrix-based methods) ?**
> > >
> > > **A9**: We visualize the t-SNE of the noise transition inferred by our method and MEIDTM. We also select 30 data points with the same predicted clean labels. The dataset is CIFAR-10 with instance-dependent label noise, and the noise rate is 50%. The experiment results are shown in Appendix G. The pairs of data points with the same predicted clean label are marked with the same number. The distances between two data points represent the difference between the two noise transitions of these data points.
> > >
> > > We can see that the distance between the same pair is different in the two images. For example, the pairs with number 21 are close to each other in the first figure but are further apart in the second figure. This can verify that the similarity inferred by our method is different from the instance-dependent transition matrix-based method.

---

> ### Author Response · Authors · 2023-11-22
> **Response to Reviewer XDR1 (Continue)**
>
> ### Reference
>
> [1] Hyvärinen, Aapo, and Petteri Pajunen. "Nonlinear independent component analysis: Existence and uniqueness results." *Neural networks* 12.3 (1999): 429-439.
>
> [2] Locatello, Francesco, et al. "Challenging common assumptions in the unsupervised learning of disentangled representations." *international conference on machine learning*. PMLR, 2019.
>
> [3] Yang, Mengyue, et al. "Causalvae: Disentangled representation learning via neural structural causal models." *Proceedings of the IEEE/CVF conference on computer vision and pattern recognition*. 2021.
>
> [4] Liu, Yuhang, et al. "Identifying weight-variant latent causal models." *arXiv preprint arXiv:2208.14153* (2022).
>
> [5] Xia, Xiaobo, et al. "Part-dependent label noise: Towards instance-dependent label noise." *Advances in Neural Information Processing Systems* 33 (2020): 7597-7610.
>
> [6] Xia, Xiaobo, et al. "Are anchor points really indispensable in label-noise learning?." *Advances in neural information processing systems* 32 (2019).
>
> [7] Cheng, De, et al. "Class-Dependent Label-Noise Learning with Cycle-Consistency Regularization." *Advances in Neural Information Processing Systems* 35 (2022): 11104-11116.

---

> ### Comment · Reviewer_XDR1 · 2023-11-23
> **Response to the authors**
>
> Thank the authors again for their sincere efforts to relieve my concerns. I have questions below:
>
> 1. What is the reference [6] and [7]? I think there are only 5 references...
>  If [7] is [7] Patrini, Giorgio, et al. "Making deep neural networks robust to label noise: A loss correction approach." Proceedings of the IEEE conference on computer vision and pattern recognition. 2017., what is the difference between CCR and Forward?
>
> 2. I doubt experimental results are chosen since the dataset has changed from CIFAR10-IDN to CIFAR10-N from Q5. If the authors can explain why, I can understand.
>
> 3. According to various experimental results, I can compare at least BLTM, MEIDTM and GenP with regard to accuracy, transition matrix error. For accuracy, I found out BLTM<MEIDTM<GenP, while for transition matrix error, it is GenP<BLTM<MEIDTM. I want to say that the order of transition matrix error and accuracy is reversed at least for BLTM and MEIDTM. Is it right? Also, I want to see accuracy comparison with VolMinNet.
>
> 4. Is this method applicable to practical situations? For example, if we need 30 times more time (as the authors reported in the responses) for increasing 6% accuracy (as the authors reported in Clothing1M) than the naive cross entropy, is it practical?
>
> 5. It is just my curiosity (and it may not be important for the classification accuracy, but I'm not sure). Since GenP reconstructs the image again, can authors show how the image is generated? If the method worked correctly, the image should also have been generated correctly, right? I know that we do not have enough time to run the algorithm totally according to the reported time from the authors for this revision, but I suggest showing this generated image can be a way to show the algorithm has worked as the authors intended.

---

> > ### Author Response · Authors · 2023-11-23
> > **Response to Reviewer XDR1**
> >
> > Thanks for your response.
> >
> > **Q1: What is the reference [6] and [7].**
> >
> > **A1**: We apologize for this mistake, and we have added the missing citations.
> >
> > **Q2: The dataset has changed from CIFAR10-IDN to CIFAR10-N from Q5.**
> >
> > **A2:** In our paper, the experiments detailed in Section A2 primarily focus on demonstrating the robustness of our algorithms and the effectiveness of standard cross-entropy loss across varying noise rates. For this purpose, we employed the CIFAR10-IDN dataset, which allows us to generate noisy datasets with different noise rates. This choice enables a more controlled examination of algorithm performance under noise variations.
> >
> > In Question 5, the results presented in Q5 do not directly pertain to test accuracy performance but rather to the training time and memory usage of each algorithm.
> >
> > In Questions 7 and 8, the results in the response are a subset of the experiment results in our papers. To facilitate a more direct and convenient comparison for you, we have included in this response the experimental results conducted on the real-world dataset CIFAR-10N. The full results can be found in Table 1~4 in our paper.
> >
> > **Q3: The order of transition matrix error and accuracy is reversed at least for BLTM and MEIDTM. Is it right?**
> >
> > **A3**: We have carefully checked the code and rerun it. The experiment results are consistent with the results in the paper. This phenomenon is interesting, and we will study it in the future.
> >
> > **Q4: The accuracy comparison with VolMinNet.**
> >
> > **A4**: We have conduct experiments to make comparison with VolMinNet. Due to time limitation, we only conduct the experiment in the CIFAR-10N dataset. The accuracy comparison results are shown as follows.
> >
> > |           | Worst                | Aggregate            | Random 1             | Random 2             | Random 3             |
> > | --------- | -------------------- | -------------------- | -------------------- | -------------------- | -------------------- |
> > | VolMinNet | 73.11 $\pm$ 0.35     | 88.38 $\pm$ 0.11     | 85.53 $\pm$ 0.11     | 85.15 $\pm$ 0.14     | 85.36 $\pm$ 0.17     |
> > | GenP      | **93.87 $\pm$ 0.13** | **95.39 $\pm$ 0.18** | **95.38 $\pm$ 0.13** | **95.30 $\pm$ 0.12** | **95.26 $\pm$ 0.13** |
> >
> > **Q5: Is this method applicable to practical situations?.**
> >
> > **A5**: There is a trade-off between increased training time and improved accuracy. In some real-world application scenes, it is worth increasing the training time for higher accuracy. For example, in financial services, particularly in fraud detection, a slight improvement in accuracy can mean the difference between detecting a  fraudulent transaction and missing it. Financial institutions might be willing to allocate more computational resources if it means reducing the incidence of fraud, which can have substantial financial implications.
> >
> > **Q6: The reconstructed images visualization.**
> >
> > **A6:** Thank you for your valuable suggestion. Following your advice, we have visualized the reconstructed images from our model trained on the FashionMNIST dataset with a noise rate of 0.5. Due to time constraints, we present the reconstruction results after 30 epochs of training, whereas our full experiments in the paper involve 300 epochs.
> >
> > These visualization results are detailed in Appendix J of our paper. For clarity, the images are arranged in a comparative format: the first and third columns display the original images, while the second and fourth columns show the corresponding reconstructed images by our model, alternating in this pattern throughout the display.
> >
> > Despite the model being in its early training phase, at just 30 epochs, the results are promising. The model can successfully reconstruct images. We believe that if the model is trained for 300 epochs, the image quality reconstructed by the model would be enhanced.

---

> ### Comment · Reviewer_XDR1 · 2023-11-23
>
> Thanks the authors for answering all questions that I asked sincerely and sorry for my possible misunderstanding of the authors' intension.
>
> Now I admit that the method of this paper makes instance dependent transition matrix well and it shows good performances for CIFAR-10 and CIFAR-10N dataset. The paper also shows more empirical analyses of why it works.
>
> I still worry that the necessary previous baselines may not be compared enough (e.g. the authors could not state the reason of the superiority of CCR  than their method for Clothing1M during the revision period) and question its applicability to practical settings, regarding time and computation complexity issue.
>
> I hope the authors could reorganize including the analyses done during in this revision period well and upgrade their study and their manuscript for the next conference.
>
> Currently, I will keep my score as before, but I think this paper is very interesting and promising.

---

### Official Review · Reviewer_5Lfr · 2023-10-25

**Soundness:** 3 good
**Presentation:** 3 good
**Contribution:** 3 good
**Rating:** 6
**Confidence:** 4

**Summary:**

To deal with noisy labels, this paper models the generation process of noisy data using a learnable graphical model to understand the underlying causal relations, instead of directly defining and utilizing similarity of noise transitions across various instances in previous work. Experiments on various dataset with different types of label noise validate its effectiveness.

**Strengths:**

1.This paper gives an insight into the transition matrix that the instances have similar transition matrix only if the causal factors causing the label noise are similar, which does make sense.

2.Experimental results under various settings show the effectiveness of the approach.

**Weaknesses:**

1.The idea of modeling the generation process of label noise via exploiting the underlying causal relations has been proposed in [1]. Hence, it is suggested to highlight the differences between the two works in Related Work.

2.Figure 2 should be polished up. For example, in Figure 2, the difference between the blue arrow and black arrow should be explained. Meanwhile, the generative process related to Figure 2 in Section 3, the core of this paper, should be give more details about.

3.The assumption that the generation process of causal factors Z is linear seems a bit unreasonable, which should be give more details on.

4.Eq.(1) is very confusing. It seems that $Z_i$ is the element in $\mathbf{Z}$ but also calculated by $\mathbf{Z}$, which provides an implicit constraint.

5.Some related works such as [2, 3] are missing.

[1] Yao, Yu, et al. "Instance-dependent label-noise learning under a structural causal model." NIPS, 2021.

[2] Sheng Liu, et al. “Early-Learning Regularization Prevents Memorization of Noisy Labels.” NIPS, 2020.

[3] Sheng Liu, et al. “Robust Training under Label Noise by Over-parameterization.” ICML, 2022.

**Questions:**

1. What is the meaning of $N_i$? In Section 3, it mentions that $N_i$ is the corresponding latent noise varibable. Could you take Figure 1 as an example to illustrate it?

---

> ### Author Response · Authors · 2023-11-18
> **Response to Reviewer 5Lfr**
>
> We sincerely thank you for your time and effort in reviewing our manuscript and providing valuable comments. Please find the response to your comments below.
>
> **Q1: Highlight the differences from CausalNL in Related Work.**
>
> **A1**: Our method is different from CausalNL from several points.
>
> - The aim of CausalNL is to exploit the information contained in the distribution of instances to help the learning of classifiers. To achieve this, the generative model is introduced. However, **existing methods do not model the causal structure among latent variables. They only “partially” model the latent noisy data generative process without identifiablity analysis.**
>
> - Our aim is to enable deep neural networks to capture the connections of noise transitions among different instances automatically. Since once the connections are captured, the noise transition learned in some examples can be generalized to other examples. To achieve this, we propose a principled way to "fully" model the data generative process. **Our method also models the causal structure among latent variables. We also analyzed sufficient conditions required for identifiability.**
>
> Here we would like to include more details. Due to limited space, we include the details in Appendix D to avoid confusion.
>
> > CausalNL helps the learning of classifiers by exploiting the information contained in the distribution of instances. To be specific, when the latent clean label $Y$ is a cause of the instance $\mathbf{X}$, the distribution of instance $p(\mathbf{X})$ will generally contain some information about the distribution of clean class posterior $p(Y|\mathbf{X})$. To exploit the information contained in the instances, CausalNL uses the generative model to model the generative relationship between the latent variable $\mathbf{Z}$ and the observed instance $\mathbf{X}$. However, they do not model the causal structure among the latent variable $\mathbf{Z}$. Therefore, their methods only “partially” model the noisy data generative process. These methods do not analyze the identifiability of the generative process.
>
> > Our aim is to enable deep neural networks to capture the connections of noise transitions among different instances automatically. Since once the connections are captured, the noise transition learned in some examples can be generalized to other examples. To achieve this, we need to model the joint distribution of all variables $p(\mathbf{X},\tilde{Y},Y,\mathbf{Z},\mathbf{N})$. The noise transition can be obtained through
>
> $$
> p(\tilde{Y}|\mathbf{X},Y)=\frac{\int_{\mathbf{Z},\mathbf{N}}p(\mathbf{X},\tilde{Y},Y,\mathbf{Z},\mathbf{N})d\mathbf{Z}d\mathbf{N}}{\int_{\tilde{Y},\mathbf{Z},\mathbf{N}}p(\mathbf{X},\tilde{Y},Y,\mathbf{Z},\mathbf{N})d\tilde{Y}d\mathbf{Z}d\mathbf{N}}.
> $$
>
> > To obtain this joint distribution, we propose a principled way to model the whole data generative process. Specifically, we not only model the generative process from the latent variable $\mathbf{Z}$ to the instance $\mathbf{X}$ but also model and learn causal structure among different latent variables $\{Z_1,Z_2,\dots,Z_m\}$ from observed noisy data.
>
> **Q2: Difference between the blue arrow and black arrow in Fig 2.**
>
> **A2**: Thanks for your suggestion. The black arrow indicates the causal direction in the structural causal model, while the blue arrow indicates the weights are changed across the clean label $Y$. More details about the generative process are included in Sec 3.
>
> **Q3: The assumption that the generation process of causal factors $\mathbf{Z}$ is linear should be given more details.**
>
> **A3**: Since causal factor $\mathbf{Z}$ is latent, discovering the causal structure is a challenging problem. If $\mathbf{Z}$ is observed, nonlinear causal mechanisms can be discovered easily. However, we do not assume the causal factors are observed. To provide a theoretical guarantee, we assume the causal mechanism among causal factors is linear. We empirically find that the performance of the proposed method is good, which indicates that this simplicity does not have negative effects.
>
> **Q4: Eq.(1) is very confusing.**
>
> **A4**: $\mathbf{W}$ is an upper triangular matrix with zero-value diagonal elements. When calculating $Z_i$, only the parents nodes of $Z_i$ are used, i.e., $Z_i=\sum_{j\in pa_i}W_{i,j}Z_{j}$.
>
> **Q5: Some related works are missing.**
>
> **A5**: Thanks for your advice. We have added these works to the Related Work.
>
> **Q6: What is the meaning of $N_i$.**
>
> **A6**: $N_i$ is an unexplained random variable that determines $Z$ in the structural causal model, i.e., $Z_i:=f_Z(pa(Z_i),N_i)$. This variable is referred as the noise variable in previous work [1], which is different from the noise in the label noise.
>
> ### References
>
> [1] Schölkopf, Bernhard, et al. "Toward causal representation learning." *Proceedings of the IEEE* 109.5 (2021): 612-634.

---

> > ### Comment · Reviewer_5Lfr · 2023-11-21
> >
> > The authors have adequately addressed the majority of my concerns. I hope that the authors will further revise the manuscript in future versions based on the modification suggestions I have provided, especially regarding Point 3 and 4.

---

> > > ### Author Response · Authors · 2023-11-21
> > > **Thanks**
> > >
> > > Dear Reviewer 5Lfr
> > >
> > > Thank you for your constructive comments. We will carefully revise our manuscript to avoid any confusion.
> > >
> > > Warm regards,
> > >
> > > Authors

---

### Official Review · Reviewer_qcRj · 2023-11-01

**Soundness:** 3 good
**Presentation:** 3 good
**Contribution:** 2 fair
**Rating:** 5
**Confidence:** 4

**Summary:**

This paper addresses the challenge of learning with noisy labels, particularly focusing on the critical aspect of understanding noise transitions—the process by which a clean label turns into a noisy one. Traditionally, most methods infer noise transitions based on assumptions about similarities across different instances. However, these assumptions often lack empirical backing and may not accurately reflect real-world data scenarios, leading to incorrect interpretations of both noise transitions and clean labels.

To overcome these limitations, the authors propose a novel approach that models the generative process of noisy data instead of relying on predefined similarity assumptions. This method uses a learnable graphical model to represent the causal generative process behind the noise in data. By doing this, the model can more effectively identify the underlying causal factors that lead to noise in labels.

**Strengths:**

The author's effective construction of a causal graph that aligns well with the research's motivation, as well as their adept development of the graphical model and Evidence Lower Bound (ELBO), signifies a robust approach in addressing the research problem.

**Weaknesses:**

Indeed, in the field of noisy label classification, using deep generative models (DGMs) to infer latent true labels is not a novel approach. Various studies have explored this concept, leveraging the capacity of DGMs to model complex data distributions and underlying noise patterns, including [1].

[1] Noisy Prediction Calibration via Generative Model, ICML2022

From a critical standpoint, this study distinguishes itself from the ICML 2022 paper "[1]" by proposing a graphical model that requires the generation of input instances. A significant advantage of the approach in "[1]" is that it does not require input generation. Generating high-resolution instances can be inherently challenging or demand computationally intensive models like diffusion models, making the combination of input instance generation with noisy label classification potentially impractical.

The primary focus of this study on low-resolution datasets such as MNIST and CIFAR-10 in their experimental evaluations is possibly due to these inherent difficulties. Although they report results for the higher-resolution Clothing1M dataset, the lack of specific mention of standard deviations casts doubt on the reliability of these findings. Furthermore, the performance improvement over DivideMix is marginal, raising questions about the necessity of employing a deep generative model for noisy label classification. This aspect warrants skepticism regarding the scalability and practical applicability of the proposed method, especially for higher-resolution, real-world datasets.

Incorporating a deep generative model in scenarios primarily focused on classification tasks can introduce a significant computational burden. The author should compare the increased computational requirements of this additional modeling with existing baselines to provide a clearer perspective on its practical feasibility. This comparison is crucial for assessing the trade-offs between the potential benefits of improved noise handling and the increased computational demands, particularly for applications where resources are limited or efficiency is a critical factor.

**Questions:**

Q1. The use of "distillation" in the paper seems unclear, as it traditionally refers to transferring knowledge from a complex to a simpler model. If this process is not evident in the methodology, the term may be inaccurately applied. The author should clarify or reconsider its use. Using a simply trained network initially does not necessarily constitute distillation, which typically involves transferring knowledge from a more complex model.

Q2. The authors should specify how their use of a DGM framework with causal graphs and inference differs in motivation and application from the work done in CausalNL, detailing the distinct aspects of their approach.

Q3. In the context of noisy label classification, it's not typical to specifically categorize noise types as "Worst," "Aggregate," "Random 1," "Random 2," and "Random 3" without clear definitions. These terms are not standard in the literature and require clarification for proper understanding.

Q4. As previously discussed, training a Deep Generative Model (DGM) for this task may impose a significant computational burden. In this context, could the methodology benefit from incorporating pre-trained DGMs to alleviate these computational demands?

---

> ### Author Response · Authors · 2023-11-18
> **Response to Reviewer qcRj**
>
> We sincerely thank you for your time and effort in reviewing our manuscript and providing valuable comments. Please find the response to your comments below.
>
> **Q1: Various studies have explored leveraging deep generative models to model complex data distributions and underlying noise patterns, thus using deep generative models (DGMs) to infer latent true labels is not a novel approach.**
>
> **A1**: Though some previous studies also use generative models, our work is different from previous work in several points.
>
> - The aim of previous work CausalNL and InstanceGM is to exploit the information contained in the distribution of instances to help the learning of classifiers. To achieve this, the generative model is introduced. However, **existing methods do not model the causal structure among latent variables. They only “partially” model the latent noisy data generative process without identifiablity analysis.**
>
> - Our aim is to enable deep neural networks to capture the connections of noise transitions among different instances automatically. Since once the connections are captured, the noise transition learned in some examples can be generalized to other examples. To achieve this, we propose a principled way to "fully" model the data generative process. **Our method also models the causal structure among latent variables. We also analyzed sufficient conditions required for identifiability.**
>
> Here we would like to include more details. We have also included below in Appendix D to avoid confusion.
>
> > Previous work helps the learning of classifiers by exploiting the information contained in the distribution of instances. To be specific, when the latent clean label $Y$ is a cause of the instance $\mathbf{X}$, the distribution of instance $p(\mathbf{X})$ will generally contain some information about the distribution of clean class posterior $p(Y|\mathbf{X})$. To exploit the information contained in the instances, CausalNL and InstanceGM use the generative model to model the generative relationship between the latent variable $\mathbf{Z}$ and the observed instance $\mathbf{X}$. However, they do not model the causal structure among the latent variable $\mathbf{Z}$. Therefore, their methods only “partially” model the noisy data generative process. These methods do not analyze the identifiability of the generative process.
>
> > Our aim is to enable deep neural networks to capture the connections of noise transitions among different instances automatically. Since once the connections are captured, the noise transition learned in some examples can be generalized to other examples. To achieve this, we need to model the joint distribution of all variables $p(\mathbf{X},\tilde{Y},Y,\mathbf{Z},\mathbf{N})$. The noise transition can be obtained through
>
> $$
> p(\tilde{Y}|\mathbf{X},Y)=\frac{\int_{\mathbf{Z},\mathbf{N}}p(\mathbf{X},\tilde{Y},Y,\mathbf{Z},\mathbf{N})d\mathbf{Z}d\mathbf{N}}{\int_{\tilde{Y},\mathbf{Z},\mathbf{N}}p(\mathbf{X},\tilde{Y},Y,\mathbf{Z},\mathbf{N})d\tilde{Y}d\mathbf{Z}d\mathbf{N}}.
> $$
>
> > To obtain this joint distribution, we propose a principled way to model the whole data generative process. Specifically, we not only model the generative process from the latent variable $\mathbf{Z}$ to the instance $\mathbf{X}$ but also model and learn causal structure among different latent variables $\{Z_1,Z_2,\dots,Z_m\}$ from observed noisy data.
>
> To achieve this, the generation of the variable $\mathbf{N}$ is modeled by a Gaussian distribution. The generation of the latent variable $\mathbf{Z}$ is modeled by a Structure Causal Model (SCM): $Z_i:=\mathbf{W}^T_i \mathbf{Z}+N_i$, where $\mathbf{W}$ is a matrix used to represent the association among the latent variables. Two generators are used to model the generation of the instance $\mathbf{X}$ and the noisy label $\tilde{Y}$, respectively.
>
> Moreover, our paper has analyzed the identifiability of the proposed method theoretically in Appendix C. We discuss the conditions required for identifiability. Specifically, when the number of distinct classes is large (more than the sufficient statistics of the latent variable $\mathbf{Z}$) and we have a confident example in each class, the latent variable $\mathbf{Z}$ can be identified up to permutation.
>
> **Q2: The deep generative model introduces a significant computational burden.**
>
> **A2**: The proposed method does not increase the time complexity and memory because the encoder and the decoder introduced by our method are lightweight. The training time of the proposed method on CIFAR-10 is 13.68 hours, and the training time of DivideMix is 9.08 hours. The memory used for the proposed method is 7.06 GB, and the memory used for DivideMix is 6.99 GB.
>
> Moreover, the inference time is the same as the time of baselines because the encoder and decoder can be discarded during inference.

---

> > ### Author Response · Authors · 2023-11-18
> > **Response to Reviewer qcRj (Continue)**
> >
> > **Q3: The performance improvement over DivideMix is marginal, raising the question the necessity of employing a deep generative model for noisy label classification.**
> >
> > **A3**: Performance improvement is dramatic when the noise rate is high. For example, when the noise rate is 50%, our method improves accuracy by 3.21% on FashionMNIST, 2.41% on CIFAR-10, and 4.58% on CIFAR-100. When the noise rate is low, the performance of baselines and the proposed method is relatively good. Thus, the improvement is marginal.
> >
> > **Q4: Could the methodology benefit from incorporating pre-trained DGMs to alleviate these computational demands?**
> >
> > **A4**: Thank you for the interesting question. No, our goal cannot be achieved if we use pre-trained DGMs. Our goal is to enable deep networks to understand the connections of noise transitions among different instances automatically (please refer to the answer in A1). To capture the connections, we need to obtain the joint distribution of all variables $p(\mathbf{X},\tilde{Y},Y,\mathbf{Z},\mathbf{N})$.
> >
> > To achieve this, the generation of the variable $\mathbf{N}$ is modeled by a Gaussian distribution. The generation of the latent variable $\mathbf{Z}$ is modeled by a Structure Causal Model (SCM): $Z_i:=\mathcal{W}^T_i \mathcal{Z}+N_i$, where $\mathcal{W}$ is a matrix used to represent the association among the latent variables. Two generators are used to model the generation of the instance $\mathbf{X}$ and the noisy label $\tilde{Y}$, respectively.
> >
> > Moreover, our paper has analyzed the identifiability of the proposed method theoretically in Appendix C. We discuss the conditions required for identifiability. Specifically, when the number of distinct classes is large (more than the sufficient statistics of the latent variable $\mathbf{Z}$) and we have a confident example in each class, the latent variable $\mathbf{Z}$ can be identified up to permutation. In contrast, existing DGMs can not provide a theoretical guarantee for inferred latent variables. The accurate joint distribution $p(\mathbf{X},\tilde{Y},Y,\mathbf{Z},\mathbf{N})$ may not be obtained. Without the accurate joint distribution $p(\mathbf{X},\tilde{Y},Y,\mathbf{Z},\mathbf{N})$, the model can not capture the connections of noise transitions among different instances. Therefore, we can not use pre-trained DGMs.
> >
> > **Q5: The word "distillation" traditionally refers to transferring knowledge from a complex to a simpler model.**
> >
> > **A5**: Thank you for pointing out. To avoid confusion, we have changed the word to "selection" in the paper.
> >
> > **Q6: It's not typical to specifically categorize noise types as noise types as "Worst," "Aggregate," "Random 1," "Random 2," and "Random 3".**
> >
> > **A6**: These noise types are not categorized by us but by the original paper [1]. Their definition can be found in the original paper [3]. Specifically, "Random $i$" ($i\in\{1,2,3\}$) represents the $i$-th submitted label, "Aggregate" represents the aggregation of three noisy labels by majority voting, and "Worst" represents the dataset with the highest noise rate.
> >
> > ### References
> >
> > [1] Wei, Jiaheng, et al. "Learning with noisy labels revisited: A study using real-world human annotations." *arXiv preprint arXiv:2110.12088* (2021).

---

> > > ### Comment · Reviewer_qcRj · 2023-11-21
> > > **Thanks for your response**
> > >
> > > I acknowledge that most concerns regarding other questions have been addressed, and I admit there was some misunderstanding on my part.
> > >
> > > However, regarding Q1, I specifically requested a direct comparison with NPC [1], but only comparisons with other DGM-based baselines were provided. I would like a detailed discussion on the differences between your approach and NPC, along with the benefits of these differences. A performance comparison would be particularly valuable.

---

> > > > ### Author Response · Authors · 2023-11-22
> > > > **Differences between our approach and NPC and comparison**
> > > >
> > > > Dear Reviewer qcRj,
> > > >
> > > > Thank you for your constructive comments.
> > > >
> > > >  **NPC is a post-processing method that can be integrated into existing noise-robust methods. On the other hand, our approach is a specific solution designed to help learn noise transitions. NPC can be combined with our method to further boost our accuracy.**
> > > >
> > > > We would like to first explain more about the major difference between methods and NPC.  Then we illustrate the experimental results by combining our method with NPC.
> > > >
> > > > - The NPC focuses on modeling the **generative processes of predicted clean labels of a trained classifier.** Therefore, it is used as a post-processing method that 1). models the predicted label generation process of all existing noise-robust methods; 2. further refine the predicted label to improve the calibration.
> > > >
> > > >
> > > > - In contrast, our method models the generative processes of noisy data.  Our aim is to enable deep neural networks to capture the connections of noise transitions among different instances automatically. Since once the connections are captured, the noise transition learned in some examples can be generalized to other examples. To achieve this, we propose a principled way to model the data-generative process. We also analyzed sufficient conditions required for identifiability. Please also refer **A1** of our rebuttal for a detailed explanation of our method.
> > > >
> > > >
> > > >
> > > > **NPC can be combined with our method to further boost the accuracy**. We have combined our method with NPC on CIFAR-10N, the results are as follows.
> > > >
> > > > |            | Worst            | Aggregate        | Random 1         | Random 2         | Random 3         |
> > > > | ---------- | ---------------- | ---------------- | ---------------- | ---------------- | ---------------- |
> > > > | GenP       | 93.87 $\pm$ 0.13 | 95.39 $\pm$ 0.18 | 95.38 $\pm$ 0.13 | 95.30 $\pm$ 0.12 | 95.26 $\pm$ 0.13 |
> > > > | GenP + NPC | 94.06 $\pm$ 0.15 | 95.48 $\pm$ 0.16 | 95.53 $\pm$ 0.14 | 95.49 $\pm$ 0.15 | 95.41 $\pm$ 0.17 |
> > > >
> > > > **We will carefully sort this out in our final version.** Thank you for pointing this out.
> > > >
> > > > Warm regards,
> > > >
> > > > Authors

---

> > > > > ### Comment · Reviewer_qcRj · 2023-11-23
> > > > > **Thanks for your response**
> > > > >
> > > > > The detailed analysis of NPC is expected to greatly aid in the analysis of the DGM-based methodology. Based on this, I will raise my score to 5.

---

> > > > > > ### Author Response · Authors · 2023-11-23
> > > > > > **Thanks**
> > > > > >
> > > > > > Reviewer qcRj,
> > > > > >
> > > > > > Thanks for your comments. Thank you again for investing time and energy in reviewing the paper.
> > > > > >
> > > > > > Best regards,
> > > > > >
> > > > > > Authors

---

### Official Review · Reviewer_5FMB · 2023-11-02

**Soundness:** 2 fair
**Presentation:** 2 fair
**Contribution:** 2 fair
**Rating:** 5
**Confidence:** 3

**Summary:**

The paper proposes a graphical modeling approach for label-noise learning. It addresses the issue of manual assumptions regarding noise transition in previous work, so they design the generative process based on causal factors. These causal factors is latent, so they propose learnable generative models to establish relationships among data instances, clean labels, and noisy labels. The proposed method, GenP, shows better performances than existing methods.

**Strengths:**

The motivation and problem formulation are reasonable. Designing a graphical model is an effective approach to uncovering the connections related to unknown transitions. The manuscript is well-written.

**Weaknesses:**

1. Missing explanation and comparison with important baselines

* The paper states that there are "no existing methods that attempt to unveil these latent noisy data generative process". However, there are previous works that have addressed the generative processes for label-noise learning [1, 2, 3], and even they use [1] for the baseline in the experiments. The paper should provide a comprehensive discussion for these baselines.

* Additionally, there is a lack of discussion regarding related work on label-noise learning. While they use the clean example distillation method via the small-loss trick, it is essential to also survey this related work, such as sample selection methods, for a more comprehensive view of the field.

[1] Yao, Y., Liu, T., Gong, M., Han, B., Niu, G., & Zhang, K. (2021). Instance-dependent label-noise learning under a structural causal model. Advances in Neural Information Processing Systems, 34, 4409-4420.

[2] Bae, H., Shin, S., Na, B., Jang, J., Song, K., & Moon, I. C. (2022, June). From noisy prediction to true label: Noisy prediction calibration via generative model. In International Conference on Machine Learning (pp. 1277-1297). PMLR.

[3] Garg, A., Nguyen, C., Felix, R., Do, T. T., & Carneiro, G. (2023). Instance-dependent noisy label learning via graphical modelling. In Proceedings of the IEEE/CVF Winter Conference on Applications of Computer Vision (pp. 2288-2298).

2. Lack of Experimental Support

* Some results, such as those for CIFAR-10 and CIFAR-10N, do not appear to be statistically significant, and the performance gain observed for Clothing1M is marginal. To further support the effectiveness of the proposed model, it would be beneficial to include experiments on various real-world datasets such as WebVision, ANIMAL-10N, and Mini-Imagenet, or to conduct repeated experiments on Clothing-1M.

* The paper primarily presents the classification performance without conducting an in-depth analysis of the contributing factors behind the performance improvements. A more detailed examination of specific elements that have positively influenced the overall performance is needed. This could include: 1) an ablation study comparing model training with only $L_{semi}$ and with the entire loss, 2) a sensitivity analysis regarding hyperparameters (e.g., $\lambda_{ELBO}$ and $\lambda_M$), 3) a comparison between end-to-end learning and alternative learning approaches, and 4) an ablation study based on the number of latent variables, among other potential analyses.

3. Minor Comments

* The first paragraph of Section 3 and the first paragraph of 'Intuition about Inferring Latent Generative Process' have significant overlap. Rephrasing these sentences would improve clarity and avoid redundancy.

* It would be helpful to include a citation when introducing the "small-loss trick" to provide proper credit and context.

* Some numbering in the 'Baselines' section appears to be missing (e.g., 5 and 11).

**Questions:**

Please answer the comments in the Weaknesses section.

---

> ### Author Response · Authors · 2023-11-18
> **Response to Reviewer 5FMB**
>
> We sincerely thank you for your time and effort in reviewing our manuscript and providing valuable comments. Please find the response to your comments below.
>
> **Q1: There is previous work that models the generative processes for label-noise learning.**
>
> **A1**: We are sorry for the confusion. We believe this is caused by a misunderstanding of noisy data generative process.
>
> - The aim of previous work CausalNL and InstanceGM is to exploit the information contained in the distribution of instances to help the learning of classifiers. To achieve this, the generative model is introduced. However, **existing methods do not model the causal structure among latent variables. They only “partially” model the latent noisy data generative process without identifiablity analysis.**
>
> - Our aim is to enable deep neural networks to capture the connections of noise transitions among different instances automatically. Since once the connections are captured, the noise transition learned in some examples can be generalized to other examples. To achieve this, we propose a principled way to "fully" model the data generative process. **Our method also models the causal structure among latent variables. We also analyzed sufficient conditions required for identifiability.**
>
> - Note that **NPC does not model the noisy data generative process but models the generative process of estimated clean labels.**
>
> Here we would like to include more details. We have also included below in Appendix D to avoid confusion.
>
> > Previous work helps the learning of classifiers by exploiting the information contained in the distribution of instances. To be specific, when the latent clean label $Y$ is a cause of the instance $\mathbf{X}$, the distribution of instance $p(\mathbf{X})$ will generally contain some information about the distribution of clean class posterior $p(Y|\mathbf{X})$. To exploit the information contained in the instances, CausalNL and InstanceGM use the generative model to model the generative relationship between the latent variable $\mathbf{Z}$ and the observed instance $\mathbf{X}$. However, they do not model the causal structure among the latent variable $\mathbf{Z}$. Therefore, their methods only “partially” model the noisy data generative process. These methods do not analyze the identifiability of the generative process.
>
> > Our aim is to enable deep neural networks to capture the connections of noise transitions among different instances automatically. Since once the connections are captured, the noise transition learned in some examples can be generalized to other examples. To achieve this, we need to model the joint distribution of all variables $p(\mathbf{X},\tilde{Y},Y,\mathbf{Z},\mathbf{N})$. The noise transition can be obtained through
>
> $$
> p(\tilde{Y}|\mathbf{X},Y)=\frac{\int_{\mathbf{Z},\mathbf{N}}p(\mathbf{X},\tilde{Y},Y,\mathbf{Z},\mathbf{N})d\mathbf{Z}d\mathbf{N}}{\int_{\tilde{Y},\mathbf{Z},\mathbf{N}}p(\mathbf{X},\tilde{Y},Y,\mathbf{Z},\mathbf{N})d\tilde{Y}d\mathbf{Z}d\mathbf{N}}.
> $$
>
> > To obtain this joint distribution, we propose a principled way to model the whole data generative process. Specifically, we not only model the generative process from the latent variable $\mathbf{Z}$ to the instance $\mathbf{X}$ but also model and learn causal structure among different latent variables $\{Z_1,Z_2,\dots,Z_m\}$ from observed noisy data.
>
> To achieve this, the generation of the variable $\mathbf{N}$ is modeled by a Gaussian distribution. The generation of the latent variable $\mathbf{Z}$ is modeled by a Structure Causal Model (SCM): $Z_i:=\mathbf{W}^T_i \mathbf{Z}+N_i$, where $\mathbf{W}$ is a matrix used to represent the association among the latent variables. Two generators are used to model the generation of the instance $\mathbf{X}$ and the noisy label $\tilde{Y}$, respectively.
>
> Moreover, our paper has analyzed the identifiability of the proposed method theoretically in Appendix C. We discuss the conditions required for identifiability. Specifically, when the number of distinct classes is large (more than the sufficient statistics of the latent variable $\mathbf{Z}$) and we have a confident example in each class, the latent variable $\mathbf{Z}$ can be identified up to permutation.

---

> ### Author Response · Authors · 2023-11-18
> **Response to Reviewer 5FMB (Continue)**
>
> **Q2: Need to discuss more related work for learning with noisy labels.**
>
> **A2**: Thanks for your advice. We have added a paragraph in related work after introducing the noise-transition-based method to introduce other approaches, including sample-selection-based methods, other methods using generative models to facilitate learning with noisy labels, and the method leveraging the property of the label noise. Specifically, the paragraph is as follows.
>
> > Some noise-robust algorithms select examples deemed likely to be accurate for training purposes. These selections are based on the memorization effect, which suggests deep neural networks initially learn dominant patterns before progressively learning less common ones. In noisy label environments, accurate labels often constitute the majority, leading networks to prioritize learning from examples with accurate labels, typically indicated by lower loss values. Co-Teaching employs this principle to identify low-loss examples as likely accurate. DivideMix uses a Gaussian Mixture Model to separate training examples into labeled and unlabeled sets based on their training loss, with the labeled set presumed to contain accurate labels. Additionally, some methods use generative models to facilitate learning with noisy labels. CausalNL and InstanceGM utilize instance-specific information to enhance classifier learning. Conversely, NPC focuses on the generative process of estimated clean labels, not the noisy data generation, using generative models for label calibration. Finally, SOP applies the sparse property of the label noise, i.e., incorrect labels are the minority, to prevent models from overfitting to label noise.
>
> **Q3: Include experiments on various real-world datasets such as WebVision, ANIMAL-10N, and Mini-Imagenet.**
>
> **A3:** Thanks for your advice. We are still running experiments on WebVision. We will post the results as soon as possible.
>
> **Q4: An ablation study and sensitivity analysis.**
>
> **A4:** Thanks for your advice. For the ablation study, the method with only $\mathcal{L}\_{semi}$ is the same as DivideMix. Thus, the experiment results of the model only trained with $L_{semi}$ are the experiment results of DivideMix. For the sensitivity analysis, we have conducted the sensitivity analysis for $\lambda_{ELBO}$ and $\lambda_M$ on the CIFAR-10N dataset with noise type ''worst''. The experiment results are plotted as figures in Appendix E. We also list the experiment results as tables here:
>
> | $\lambda_{ELBO}$ | 0.1              | 0.05             | 0.01                 | 0.001            | 0.0001           |
> | ---------------- | ---------------- | ---------------- | -------------------- | ---------------- | ---------------- |
> | Accuracy         | 93.00 $\pm$ 0.20 | 93.86 $\pm$ 0.10 | **93.87 $\pm$ 0.11** | 93.56 $\pm$ 0.15 | 93.59 $\pm$ 0.11 |
>
> | $\lambda_{M}$ | 0.2              | 0.1              | 0.01                 | 0.001            | 0.0001           |
> | ------------- | ---------------- | ---------------- | -------------------- | ---------------- | ---------------- |
> | Accuracy      | 93.11 $\pm$ 0.21 | 93.44 $\pm$ 0.22 | **93.80 $\pm$ 0.13** | 93.53 $\pm$ 0.14 | 93.23 $\pm$ 0.17 |

---

> > ### Comment · Reviewer_5FMB · 2023-11-20
> >
> > Thank you for the response and for providing additional experimental results. It addresses some of my concerns. However, after reading the author's response and the other reviewers' reviews and responses to them, I still have a number of concerns.
> >
> > In particular, I still lack evidence that the proposed method is superior, as other reviewers have pointed out. I believe that the author's statement that it is conceptually superior to other methods needs to be supported by a theoretical/experimental basis, which could be ablation studies as I suggested, or the author should have designed an experiment to demonstrate superiority.
> >
> > Therefore, I find it difficult to decide on acceptance without further analysis of the proposed model.

---

> ### Author Response · Authors · 2023-11-22
> **Additional Experiments [Part 1]**
>
> Dear Reviewer 5FMB
>
> Thank you for your valuable comments on improving our paper. We have added more experiments.  These experiment results are added to Appendix E and H. Please let us know if there are any further concerns. Many Thanks.
>
>
> 1). **The ablation study compares model training with only $\mathcal{L}_{semi}$ and with the entire loss.** Here, we show the experiment results on CIFAR-10 with instance-dependent label noise. The experimental results indicate that with the increase of noise levels, the benefits of modeling the generative process become more significant (>2% on IDN-40%).
>
> |                                | IDN-10%              | IDN-20%              | IDN-30%              | IDN-40%              | IDN-50%              |
> | ------------------------------ | -------------------- | -------------------- | -------------------- | -------------------- | -------------------- |
> | With only $\mathcal{L}_{semi}$ | 96.03 $\pm$ 0.14     | 95.92 $\pm$ 0.12     | 95.66 $\pm$ 0.15     | 95.03 $\pm$ 0.12     | 86.98 $\pm$ 0.28     |
> | The entire loss                | **96.12 $\pm$ 0.12** | **96.05 $\pm$ 0.12** | **95.74 $\pm$ 0.13** | **95.44 $\pm$ 0.12** | **89.39 $\pm$ 0.45** |
>
> *2). A sensitivity analysis regarding hyperparameters ($\lambda_{ELBO}$ and $\lambda_{M}$).*
>
> We have conducted the sensitivity analysis for $\lambda_{ELBO}$ and $\lambda_M$ on the CIFAR-10N dataset with noise type ''worst''. The experiment results are plotted as figures in Appendix E. The experiment results show that when the $\lambda_{ELBO}$ and $\lambda_M$ are around $0.01$, the test accuracy of the model is highest.

---

> ### Author Response · Authors · 2023-11-22
> **Additional Experiments [Part 2]**
>
> *3). A comparison between end-to-end learning and alternative learning approaches.*
>
> We conduct the experiments of the alternative learning approach, which optimizes $\mathcal{L}\_{semi}$ and $ - \lambda\_{ELBO} ELBO +\lambda\_M (\| \mathbf{M\_X} \|_1 + \| \mathbf{M\_{\tilde{Y}}} \|\_1)$ alternatively. The dataset is CIFAR-10N.
>
> The experiment results are shown as follows. Empirically, the performance of the end-to-end learning approach is better than the alternative learning approach.
>
> |             | Worst                | Aggregate            | Random 1             | Random 2             | Random 3             |
> | ----------- | -------------------- | -------------------- | -------------------- | -------------------- | -------------------- |
> | Alternative | 93.72 $\pm$ 0.07     | 95.21 $\pm$ 0.19     | 95.14 $\pm$ 0.12     | 95.28 $\pm$ 0.09     | 94.99 $\pm$ 0.15     |
> | End-to-end  | **93.87 $\pm$ 0.13** | **95.39 $\pm$ 0.18** | **95.38 $\pm$ 0.13** | **95.30 $\pm$ 0.12** | **95.26 $\pm$ 0.13** |
>
> *4). An ablation study based on the number of latent variables.*
>
> We have conducted the experiments on CIFAR-10N dataset with the noise type "worst". The experiment results are in Appendix H. When the number of latent variables is around 4, the classification performance of the model reaches the peak.

---

> > ### Comment · Reviewer_5FMB · 2023-11-23
> >
> > Thank you for providing the ablation studies. I will consider for the response with other reviewers.
> >
> > Additionally, I think that the results for the first ablation study come from the Table 2 in the manuscript (so, I think these results are on CIFAR-10, not CIFAR-10N.), as the authors said that the method with only $\mathcal{L}_{semi}$ is the same as DivideMix. I find that the accuracy of DividMix reported in this paper and in BLTM (Table 7) is quite different (e.g. 96.03 vs 83.31 in IDN-10%). I would like to know the reason, because this difference could affect GenP, since DivideMix is an ablation of GenP.

---

> > > ### Author Response · Authors · 2023-11-23
> > >
> > > Dear Reviewer 5FMB,
> > >
> > > Thank you for your comments. Our first ablation study was carried out on the CIFAR-10 dataset. Regarding the observed discrepancies in the accuracy of DivideMix as reported in the BLTM study compared to our results, the exact cause of these significant differences is unclear. We speculate that this variation may stem from differences in experimental setups. In our experiments, the experimental settings are the same as those in the DivideMix. This consistency in the experimental setting can ensure the comparability of the results of GenP and DivideMix.
> > >
> > > Best regards,
> > >
> > > Authors

---

### Meta-Review · Area_Chair_KN8K · 2023-12-02

**Metareview:**

Reviewers raised a number of concerns around the comparison to existing baselines, both conceptually and empirically. The author response clarified some points, and included a large number of new results. Based on these, reviewers generally had a more favorable impression of the work, but the consensus view was still that it falls below the acceptance threshold (e.g., with some discrepancies in the empirical results compared to prior works not being fully explained). It is still recommended that the authors incorporate the multiple suggestions and changes into an updated manuscript, which could be subject to a fresh round of reviews.

**Justification For Why Not Higher Score:**

Multiple concerns around comparison to existing work were raised in original reviews. Response included a large number of new results which partially alleviates these. However, with some concerns remaining, and the number of new results leading to a significantly altered manuscript, it is difficult to make a case for publication in its present form.

**Justification For Why Not Lower Score:**

N/A

---

### Decision · Program_Chairs · 2024-01-16

Reject